# Planning in a recurrent neural network that plays Sokoban

## Abstract

How a neural network (NN) generalizes to novel situations depends on whether it has learned to select actions heuristically or via a planning process. Guez et al. (2019, "An investigation of model-free planning") found that a recurrent NN (RNN) trained to play Sokoban appears to plan, with extra computation steps improving the RNN's success rate. We replicate and expand on their behavioral analysis, finding the RNN learns to give itself extra computation steps in complex situations by "pacing" in cycles. Moreover, we train linear probes that predict the future actions taken by the network and find that intervening on the hidden state using these probes controls the agent's subsequent actions. Leveraging these insights, we perform model surgery, enabling the convolutional NN to generalize beyond its $10 \times 10$ architectural limit to arbitrarily sized inputs. The resulting model solves challenging, highly off-distribution levels. We open-source our model and code, and believe the neural network's small size (1.29M parameters) makes it an excellent model organism to deepen our understanding of learned planning.

## 1 Introduction

In many tasks, the performance of both humans and some neural networks (NNs) improves with more reasoning: whether by giving a human time to think before making a chess move, or by prompting or training a large language model (LLM) to reason step by step (Kojima et al., 2022; OpenAI, 2024).

Among other reasoning capabilities, goal-oriented reasoning is particularly relevant to AI alignment. So-called "mesa-optimizers" – AIs that have learned to pursue goals through internal reasoning (Hubinger et al., 2019) – may internalize goals different from the training objective, leading to harmful misgeneralization (Di Langosco et al., 2022; Shah et al., 2022). Understanding how popular NNs without a special inductive bias learn to plan and represent the planning objective could be key to detect, prevent or correct goal misgeneralization.

Within goal-oriented reasoning, we distinguish between *plans* and *search algorithms*. A *plan* is an internal representation of a sequence of the actions which an agent will take. A *search algorithm* is an algorithm that considers many plans, evaluates them according to their predicted outcomes, and picks the plan evaluated as best. Search algorithms perform better the more plans they can evaluate and, with enough compute, generalize very well to novel problem instances (Russell & Norvig, 2009).

In this work, we investigate search and planning in a Deep Recurrent ConvLSTM (DRC) that plays Sokoban, a challenging puzzle game that remains a benchmark for planning algorithms (Peters et al., 2023) and reinforcement learning (Chung et al., 2024). Guez et al. (2019) formulated the DRC, an architecture without any special inductive bias towards planning. Despite that, they argue that it performs search internally, because 1) the DRC performs unusually well in procedural environments, 2) it generalizes well from a small variety of levels, and 3) it solves about 5% more levels with extra compute at test time.

### 1.1 Contributions

The main contribution of this work is providing conclusive evidence that the DRC internally represents a plan. Firstly, we **find where the plan is represented in the activations** using linear probes (classifiers trained on NN activations) like concurrent work (Anonymous, 2025). Secondly, we show

that intervening on the probe logits to change the plan causes the agent to execute the new plan. In other words, the plan representation is **causal**.

We provide evidence that the DRC **improves its plan with more computation**, and that it has learned to use this capability when it needs to, and not just when forced by no-ops. Firstly, we observe that the length of the plan and its accuracy at predicting future actions both increase over time. Secondly, we observe that the DRC sometimes takes actions which return to previous states of the environment, in effect wasting time. This overwhelmingly happens at the beginning of episodes and can be substituted by no-ops. Additionally, the average per-step increase in plan length and improvement in plan F1-score is larger during cycles than during non-cycles, even controlling for cycles being at the beginning of episodes. Altogether this suggests that utilizing additional computation to improve the plan (e.g. by creating state cycles) is part of the network's behavior and not an accident.

Thirdly, we find several pieces of evidence which add weight to the hypothesis that the DRC agent is performing a *search algorithm* of some kind. However, we cannot rule out the DRC using heuristics that produce plans and take multiple steps to compute. We find behaviors that are more likely if the DRC is doing search: Giving the RNN time to think with no-ops (Guez et al., 2019) disproportionately helps with levels that are hard (fig. 1, bottom-left) and have longer solutions (fig. 5, right), or that require waiting many steps before the first reward (fig. 5, left). We can use the representations in the DRC's convolutions to **generalize beyond the** $10 \times 10$ **grids in the training dataset** to larger, more challenging levels (fig. 2, right). This illustrates that the DRC's representations can generalize out-of-distribution very well and, as Guez et al. (2019) argued, search generalizes very well in procedural environments.

Finally, we fully open-source all training and interpretability data, tools, and trained models, unlike previous work. We believe the DRC is an excellent model organism for understanding planning in sophisticated neural networks. Its behavior is complex enough to be interesting and its size (1.29M parameters) is small enough to be tractable to reverse engineer. We hope that future work can use these open-source resources to find conclusive evidence of *search* and understand how NNs learn to implement it.

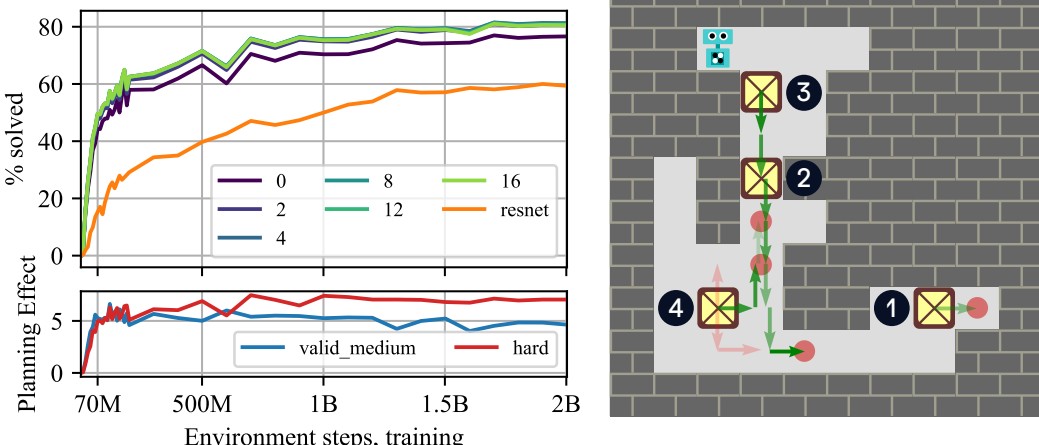

Figure 1: **Top left:** Proportion of medium-difficulty validation levels solved vs. environment steps used in training. Curves show the DRC's performance with a specific number of thinking steps (forced no-op actions) at episode start, along with a ResNet baseline. **Bottom left:** Estimated planning effect: the 8-steps minus the 0-steps curve. We see that planning emerges in the first 70M steps and keeps increasing for the hard levels (red), but decreases for medium levels (blue). **Right:** Linear probe predictions (arrows) for the direction the box will move in. Opacity is proportional to the number of steps in the episode in which the probe predicted a particular arrow. Correct predictions are in green and incorrect in red, the red predictions show other plausible plans for Box 4. This probe causally affects the actions of the agent, as described in section 4. Boxes are numbered in the order in which they arrive at their final target.

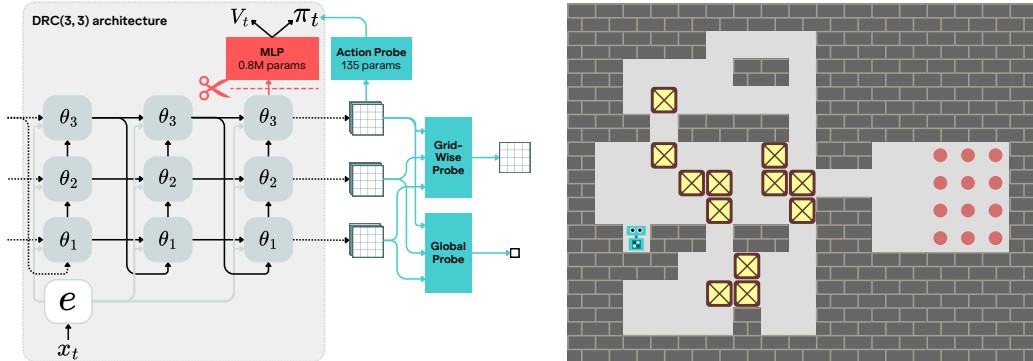

Figure 2: **Left:** The DRC$(3, 3)$ architecture from Guez et al. (2019): 3 layers of convolutions with an LSTM structure are repeated 3. The embedded observation feeds into every layer, and the last layer's output feeds into the first. *Blue:* we train two types of linear probes on the grid-cells of the 3D recurrent hidden states of the network (section 4). *Red:* The DRC$(3, 3)$ is limited to $10 \times 10$ grids by a fixed-dimension MLP. By replacing the MLP block with simple linear probes to predict action, the ConvLSTM backbone generalizes to much more challenging settings than the $10 \times 10$ training set grids (section 6.2). **Right:** XSokoban-31, which the DRC$(3, 3)$ solves after replacing the MLP.

## 2   SETTING UP THE TEST SUBJECT

We train an agent closely following the setup from Guez et al. (2019), using the IMPALA (Espeholt et al., 2018) reinforcement learning (RL) algorithm with Guez et al.'s Deep Repeating ConvLSTM (DRC) recurrent architecture. We also train a ResNet baseline. We open-source both the trained networks and training code [1] . For more details about architectures and training, see appendix A.

**DRC$(D, N)$ architecture.**   Most of this paper focuses on the behavior and representations of a DRC$(3, 3)$ neural network (Guez et al., 2019). The core of this network is a $D$-layer ConvLSTM (Shi et al., 2015), which is repeatedly applied $N$ times per environment step (fig. 2, left). The output of the last ConvLSTM layer (the $D$th layer) is fed to the input at the next tick, effectively giving the network $D \cdot N$ layers of sequential computation to output the next action. We take $D = 3, N = 3$.

A linear combination of the mean- and max-pooled activations of the ConvLSTM is also fed to the next step, thus letting it communicate quickly across the receptive field (known as *pool-and-inject*). An encoder block consisting of two $4 \times 4$ convolutions processes the input, which is also fed to each ConvLSTM layer. At the end, an MLP with 256 hidden units transforms the flattened ConvLSTM outputs into the policy (actor) and value function (critic) heads.

**Dataset.**   Sokoban is a grid puzzle game with walls, floors, movable boxes, and target tiles. The player's goal is to push all boxes onto target tiles while navigating walls. We use the Boxoban dataset (Guez et al., 2018), consisting of $10 \times 10$ procedurally generated Sokoban levels, each with 4 boxes and targets. The edge tiles are always walls, so the playable area is $8 \times 8$. Boxoban separates levels into train, validation and test sets, with three difficulty levels: unfiltered, medium, and hard. Guez et al. (2019) generated these sets by filtering levels that cannot be solved by progressively better-trained DRC networks, so easier sets occasionally contain difficult levels. In this paper, we use the unfiltered-train (900k levels) set to train networks. To evaluate them, we use the unfiltered-test (1k levels)[2], medium-validation (50k levels), and hard ($\sim$3.4k levels) sets, which do not overlap. To test DRC$(3, 3)$ generalization to different sizes, we use the levels collected by Þorsteinsson (2009) (see appendix B).

**Environment.**   The observations are $10 \times 10$ RGB images, normalized by dividing each component by 255. Each type of tile is represented by a pixel of a different color (Schrader, 2018), an example is

---
[1]URL references removed during double-blind review.

[2]We use unfiltered-test rather than unfiltered-validation so the numbers are directly comparable to Guez et al. (2019).

in fig. 12. The player can move in cardinal directions (Up, Down, Left, Right). The reward is -0.1 per step, 1 for placing a box on a target, -1 for taking it off, and 10 for finishing the level by placing all of the boxes. The time limit for evaluation is 120 steps, though large levels in section 6.2 use 1000 steps.

## 3 HYPOTHESES AND TOOLS

### 3.1 TOOLS

**Probes.** In the interpretability literature, a probe is a simple (usually linear) model which is trained to predict some label (often called the 'concept') based on intermediate activations of a NN (Alain & Bengio, 2016; Belinkov, 2016). If a simple model is able to predict a relatively complex concept, it must mean that those activations represent this concept in a very accessible way. The accuracy of a probe on a dataset is exactly the conditional $\mathcal{V}$-entropy in the theory of usable information (Xu et al., 2020), where $\mathcal{V}$ is the model class of the probe.

In this paper we train two kinds of probes (fig. 2, left), both in the model class of logistic regression: grid-wise probes (the input are channels in a single location, one probe output per grid position) and global probes (the input is the entire layer's activations). See section 4.1.

**Checking probe causality.** Probes can tell you whether a model is computing some representation, but not whether it is used for further processing. Probes often predict labels which are related to the representations the model uses to compute further steps, but which are not the representations themselves. This is more of a problem the more complex the model class $\mathcal{V}$ is.

Li et al. (2023) propose a solution to check whether their nonlinear probes are capturing what the model uses in its computation. During a forward pass of the NN, set the activations to a value that would make the output of the probe something different, and check whether the NN behavior changes accordingly.

### 3.2 HYPOTHESES: DEFINING PLANNING AND SEARCH

Guez et al. (2019) presented behavioral evidence that the DRC internally executes a planning algorithm, but they did not make precise what this means exactly. To be concrete, we distinguish between two types of (related) planning-like algorithms, and give their informal definitions here.

**Definition 1** (Plan). *A plan of a NN is a collection of future actions $\{a_t \text{ for all } t > t_0\}$, that is causally represented in the NN activations $z_{t_0}$ at time $t_0$.*

**Definition 2** (Causally represented). *Given a simple model classes for plans and whether they are selected $\mathcal{V}_p, \mathcal{V}_s$, a plan is* causally represented *if*

1. *The plan $\{a_t\}_{t>t_0}$ can be extracted from the current NN state $z_{t_0}$ using a model $v \in \mathcal{V}$.*

2. *If this plan is selected, its sequence of actions $\{a_t\}_{t>t_0}$ predicts what the NN actually does with high precision and recall.*

3. *Modifying $z_{t_0}$ changes the realized actions of the NN (future $a_t$ when running the policy) in a manner consistent with the predictive model $v$.*

**Definition 3** (Search). *A search algorithm is an algorithm that, for at least some time steps, includes the following steps: 1) generate several possible plans, 2) evaluate the value of each plan according to an approximation of its consequences (with a model of the world or otherwise), and 3) pick the plan with the best value.*

An algorithm does not need to evaluate every single intermediate state to be search, nor does its evaluation need to be perfect. Many conceivable ways of evaluating plans only use parts of a model of the world, or use heuristics to evaluate the plans. The important characteristic of *search* here is the explicit representation and choice between several possible plans.

Using these definitions, we can write hypotheses that make increasingly concrete claims.

**Hypothesis 1** (The DRC(3,3) has a plan). *The actions of the DRC(3,3) are determined by a plan that is always selected ($\mathcal{V}_p = logistic regression, \mathcal{V}_s = \{1\}$). This plan predicts and causes the actual sequence of actions the DRC takes.*

**Hypothesis 2** (Plan improves with computation). *On average, each iteration of the DRC$(3, 3)$ brings the plan closer to the actual sequence of actions the DRC executes.*

**Hypothesis 3** (Pacing to improve plans). *Sometimes, the DRC$(3, 3)$ takes more steps than would be optimal to push a box. When that happens, it is usually so that the DRC has computation to come up with a plan that solves the level, rather than because it is an incompetent agent.*

We generated hypothesis 3 from anecdotal observations of DRC behavior. It makes sense to think before box pushes because they are potentially irreversible, and even if reversible one box push takes 4 steps to correct (3 to go around the box and 1 to push it back).

This last hypothesis we do not test for directly. Instead, we provide circumstantial evidence for it in the same manner as Guez et al. (2019).

**Hypothesis 4.** *The NN is doing search: it has several plans, and selects among them based on their value.*

## 4 THE DRC$(3, 3)$ CAUSALLY REPRESENTS ITS PLAN

This section tests hypothesis 1. We present probes which predict the future actions of the DRC$(3, 3)$ and other various future features of the environment. Of these, the one which predicts the future direction *boxes* will move in has a strong *causal effect* in the actions of the DRC. Following the standard of evidence in Li et al. (2023), we declare this conclusive evidence that the DRC$(3, 3)$ represents and uses plans. The spatial structure of the probes and some of the targets are due to Anonymous (2025).

By intervening on the probes, we are able to lock the DRC into a plan when there are two options of equal value. However in most cases, if we stop intervening on the RNN activations with a suboptimal plan, the DRC comes up with a better plan online and follows it. We have some evidence for the DRC$(3, 3)$ considering multiple plans: for example, in the level in fig. 1 (right) the box probe first considers moving box 4 down and right (shown in red), but then rethinks that and ends up taking different actions. However, we have been unable to find multiple *simultaneous* representations of plans, or anything that would decide between them such as the value of the plans. In all, this section provides very little evidence in favor of *search*, but conclusive evidence for a causal *plan*.

### 4.1 PROBE METHODOLOGY: TRAINING, INTERVENTION AND TARGETS

**Training.** We train logistic or linear regression probes with L1 decay to predict various features of the environment from the agent's activations. We train on two types of inputs. **1) Grid-wise inputs** (Anonymous, 2025): each square in the $10 \times 10$ grid is a different data point, potentially with a different label depending on location and timestep. The input is the 64-dimensional LSTM state $(h, c)$ at a square, usually concatenated for the 3 layers. **2) Global inputs:** Each time step produces only one data point. The probes take as input the 64 channels of all of the squares in the 10x10 grid, resulting in a 6400-dimensional input for each layer. In both cases, the per-layer activations are concatenated for probes that are trained on multiple layers.

The dataset consists of states collected by evaluating the DRC$(3, 3)$ on the hard Boxoban levels, excluding the first 5 steps of each episode because the plan is still forming. For multi-class probes, training and F1 scores are computed as one-vs-all: the presence of absence of a particular class is the probe label. We search the best learning rate and L1 decay with grid-search by evaluating the F1 on a validation split of $20\%$ of timesteps from the hard levels. We take the first 1000 medium-validation levels as our test set to report the results (table 1).

**Causal intervention.** Given a probe that is predictive of future behavior, we edit the agent's activations such that the probe predicts the behavior we desire to induce. To the extent that the agent carries out the edited behavior, we can conclude that it executes a plan encoded in the linear direction of the probe.

For a linear probe on the activations $h$ with parameter vector $p$, we intervene in order to increase the logits for a particular behavior, given by $p \cdot h$ (we omit a scalar intercept for brevity). Unlike previous work in steering vectors for LLMs (Turner et al., 2023; Rimsky et al., 2023; Li et al., 2024), we found

that simply adding a perturbation equal to a fixed multiple of $p$ caused the policy to degenerate and choose random actions. We instead add an adaptively-scaled multiple of $p$ to the activations, chosen to minimize the change in the activations while still causing the probe's logit to be greater than a desired amount $\alpha$. More precisely, we find $\arg\min_{h'} \|h' - h\|_2^2$, with the constraint that $h' \cdot p > \alpha$. The solution to this is $h' = h + \hat{p}\max(0, \alpha - (h \cdot p))$, where $\hat{p} = p/\|p\|_2^2$.

**Targets.** We choose targets that plausibly encode the steps of a plan. Most of these labels are for grid-wise probes, but the *pacing* and *value* probes are global. The direction probes are very similar to Anonymous (2025).

- **Agent-Directions probe**: This probe has 5 outputs: NV (No Visit), UP, DOWN, LEFT, RIGHT. The probe takes as input the 64-dim hidden state activations at a square $(x, y)$ at timestep $t_i$. If the agent visits the square $(x, y)$ at a future timestep $t > t_i$, then the probe predicts the corresponding direction that the agent takes from that square. For squares that are visited multiple times in the future, the direction corresponding to the first future visit is taken as the target. If a square is never visited in a future timestep, the probe predicts NV. There is one single multinomial probe, for all time steps and locations $(x, y)$.

- **Boxes-Directions probe**: This probe is similar to the above probe, except that it is trained to predict the direction in which *any* of the four boxes will move at a given square in the future. The rationale for this probe is that the boxes, and constraints on their movement given by obstacles, are the main difficulty with Sokoban puzzles.

- **Next-Box probe**: Predicts 1 on the square of the box that the agent will move next and 0 for every other square.

- **Next-Target probe**: Predicts 1 on the square of the target that the agent will put a box in next, and 0 for every other square.

- **Pacing probe**: The global label is 1 if the agent is currently in a cycle, and 0 otherwise.

- **Value probe**: The global label is the numerical value that the critic head outputs.

Table 1: Causal and predictive probe results. Confidence is one of the mean estimator percentiles $[2.5\%, 97.5\%]$, whichever is furthest from the mean, estimated using 1000 bootstrap resamples. The AVERAGE causal probe uses 24k data points for evaluation, and the BEST-CASE probe uses 8k. The data points are sampled from the first 1000 medium validation levels.

| PROBE TARGET | PREDICTIVE PROBE RESULTS (A) | | | CAUSAL PROBE RESULTS (B) | | |
|---|---|---|---|---|---|---|
| | BEST F1 | SPARSE PROBES | | | CAUSAL EFFECT | |
| | | F1 | $L_0$-NORM | $\alpha$ | AVERAGE | BEST-CASE |
| Box-directions | $86.4 \pm 0.1$ | $73.1 \pm 0.1$ | 63 | 30 | $43.7 \pm 0.6$ | $77.8 \pm 0.9$ |
| Agent-directions | $72.3 \pm 0.1$ | $61.3 \pm 0.1$ | 121 | 10 | $7.1 \pm 0.3$ | $20.7 \pm 0.7$ |
| Next box | $74.2 \pm 0.4$ | $69.7 \pm 0.5$ | 51 | 40 | $5.5 \pm 1.0$ | $15.1 \pm 2.5$ |
| Next target | $54.3 \pm 0.5$ | $44.0 \pm 0.5$ | 32 | 30 | $4.6 \pm 0.8$ | $13.2 \pm 2.0$ |
| Pacing | $31.0 \pm 1.8$ | $31.0 \pm 1.5$ | 5 | — | — | — |

## 4.2 PROBE EVALUATION AND CAUSALITY

**Probe predictive power.** Most probes are quite predictive, as seen in table 1. The exception is the pacing probe, with $F_1 = 31.0\%$, which is not much better than the constant 1 probe, which has $F_1 = 12.8\%$. This leads us to conclude that the DRC$(3, 3)$ does not represent whether it is in a cycle or not. See fig. 1 (right) and fig. 9 for visualizations of agent-direction and box-direction probes[3].

For the value function probe, we compute the fraction of variance explained $R^2$. If we train a global probe, $R^2$ is very high: $97.7\%$–$99.7\%$ depending on the layer. However, grid-wise probes obtain much worse but still passable results: $41.0\%$–$79.2\%$ depending on layer. We visually checked whether the grid-wise probe reads off the values of different plans, but could not find any such pattern.

---

[3]The supplementary material provides visualization videos of all the probes across several levels.

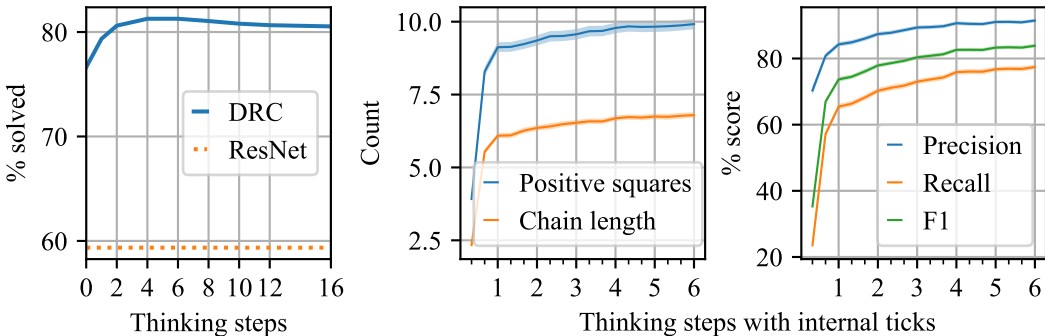

Figure 3: Success rate of the DRC, and plan quality as measured by box-direction probes (section 4.1), all increase with thinking steps. We measure plan quality by 1) summing the length of chains of probe-predicted directions, which start in boxes, and 2) simply counting the number of squares in which the box-direction probe predicts something. The F1 score of the box-direction probe also increases over time, suggesting that the DRC builds the plan in that time. The right plot also includes plan quality evaluation on the three sequential ticks per environment step of the DRC(3, 3) network.

Almost all the performance of the global probe is recovered by training on the mean-pooled inputs: 95.2%–99.5%. It is likely that the global and mean-pooled value probes are indirectly counting the number of squares the agent will step on, which almost fully determines the value, and we know is possible due to the future-direction probes.

**Only the Box-Directions probe is strongly causal.**   We measure how causal the probes are by intervening at every step on the hidden state $h, c$ at the layer the probe was trained on, following the procedure in section 4.1 and checking whether the agent follows that action. We report the result in table 1(b) by performing a grid-search over the strength of intervention $\alpha$ and picking the best value. Despite our adaptive scaling of the intervention, a high value of $\alpha$ can still disturb the agent such that it starts performing random actions whereas a low value of $\alpha$ may not be sufficient to cause the agent to follow the intended behavior. Most of the probes are not causal: only the "boxes directions" probe (used in fig. 1) is causal while the "agent-direction" probe mildly causal.

Even then, we have observed that it can only change boxes directions when they *would not deadlock* the level in a naive way. That is, if you try to make the model push a box onto a wall, such that the box would not be able to reach a target anymore, it does not. For this reason, we introduce the *Best-case causal effect* in table 1(b): try all three directions that are not the actual direction that the box follows, and count the probe as "causal" if it works for *any* of them (e.g. because they do not naively deadlock the level).

## 5   PLAN IMPROVEMENT AND PURPOSEFUL EXTRA COMPUTATION

This section tests hypotheses 2 and 3: whether the plan improves with computation and whether the DRC purposefully spends time before irreversible actions to improve it.

**Plan improvement.**   If hypothesis 2 is true, then the ability of probe-extracted plans to predict DRC actions should increase when it is given extra computation with no-ops. Figure 3 (right) shows exactly this effect: the plan starts out very incorrect, and quickly becomes much more reliable.

Figure 3 (middle) shows that plans also get more complex: the length of the plans also increases over time, at least until 6 which the probe predicts a direction grows over time.

### 5.1   AGENT "PACES" TO GET MORE COMPUTATION

On occasion, the DRC exhibits a curious behavior: the agent "paces" in a cycle, returning to the same location, without touching any box. The per-step penalty makes this behavior naively sub-optimal, because it does not advance the puzzle state.

Since the DRC$(3, 3)$'s performance improves with thinking steps, could the agent be using pacing to gain computation time to improve its plan and so avoid irreversible actions which may block the solution? The following evidence suggests the agent has developed this meta-strategy:

**The agent paces mostly at the *start* of episodes.** In some levels, a single sub-optimal step can lock the level, which makes calculating a valid plan in early steps beneficial. Figure 4 (left) shows most cycles start in the first 5 steps of the episode, and (middle) forcing the model to think for six steps eliminates about 75% of these early cycles.

**Thinking steps almost entirely substitute pacing** We run the DRC$(3, 3)$ on medium-validation levels and record when and where cycles occur. We merge all cycles which overlap with other cycles, for a total of 13 702 cycles.

We then make the DRC take $N$ thinking steps just before it would start a cycle of length $N$. As shown in fig. 4 (right), these thinking steps replace the cycles: comparing the cycle and the substitution, the DRC follows the same trajectory for at least 60% of the levels and for a minimum of 30 steps[4].

If our hypothesis that cycles are utilized for thinking holds true, we would expect to see uninterrupted periods of action after each cycle. In the $N$ steps after an $N$-step cycle concludes, 98.8% of trajectories have no cycles. When substituting them for NOOPs, this becomes 82.4%, lower but still much higher than 0%; indicating that thinking steps largely remove the NN's perceived need for cycles.

**Plan quality over time.** If the DRC uses cycles as a way to improve its plan when needed, we would expect the F1 score of plans to be lower at the beginning of cycles than during non-cycles.

We have seen that there are large effects on the F1 score of the plan from the timestep. We correct for this by pairing each cycle sample with a non-cycle sample as follows: if a cycle starts at time $t_0$, we sample another level uniformly at random, conditional on $t_0$ not being the start of a cycle. We use the hard level set. Using this protocol, the F1 score at the start of cycles is 51.42%, whereas at the start of non-cycles it is 57.80% (error bars pending but small).

We would also expect the increase in F1 score for every step to be larger in the cycles case. And it is: the F1 score increases $0.93\% \pm 0.13\%$ per step in the case of cycles, and $0.45\% \pm 0.11\%$ in the case of non-cycles. Figure 16 contains the histogram, and we can see that the distributions mostly match except the per-step improvement of cycles is right-skewed. This is some indication that 'cycles' are a bad proxy for when the DRC is in thinking mode, which also explains the failure of the pacing probe in section 4.1.

Is the higher F1 score for cycles because they are closer to the end-of-episode and have fewer things to predict? No: we can compare how many squares the plan grows by for every step of cycles and non-cycles. On average, the plan grows by 2.03 squares in the case of cycles, and 1.37 for non-cycles.

Every test that we conduct here has a small but unmistakable effect in favor of hypothesis 3.

# 6 BEHAVIORAL EVIDENCE OF SEARCH

## 6.1 EFFECTS OF THINKING TIME: NON-MYOPIC NETWORK

First, we examine the effect of thinking time: introducing no-ops at the beginning of an episode, while letting the DRC process its hidden state. As in Guez et al. (2019), adding 6 extra thinking steps lets the DRC$(3, 3)$ solve an extra 4.7% of levels, slightly decreasing when going up to 16 steps (fig. 3, left).

We find that the extra thinking time disproportionately helps with more difficult levels, as measured by longer optimal solutions (fig. 5, right). We also checked whether thinking time correlates with the number of nodes expanded by an A* search, but it does not (fig. 13).

Thinking time helps varying amounts depending on training time and level difficulties: fig. 1 (bottom left) shows that most of the planning effect steeply appears during the first 70M steps of learning.

---

[4]For context, the median solution length for train-unfiltered is exactly 30 steps.

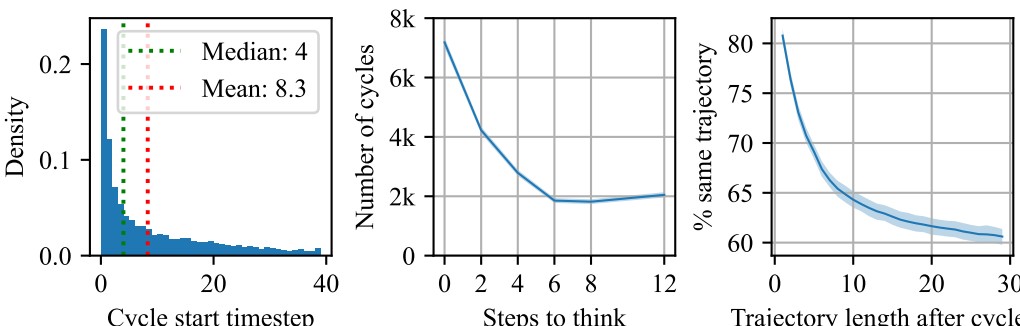

Figure 4: **Left:** Histogram of cycle start times on the medium-difficulty validation levels. **Middle:** Total number of cycles the agent takes in the first 5 steps across all episodes in medium-difficulty validation levels with $N$ initial thinking steps. **Right:** on all levels, we replace $N$-length cycles with $N$ thinking steps and check the proportion of trajectories which are equal between these two treatments, for $x$ steps after the cycle.

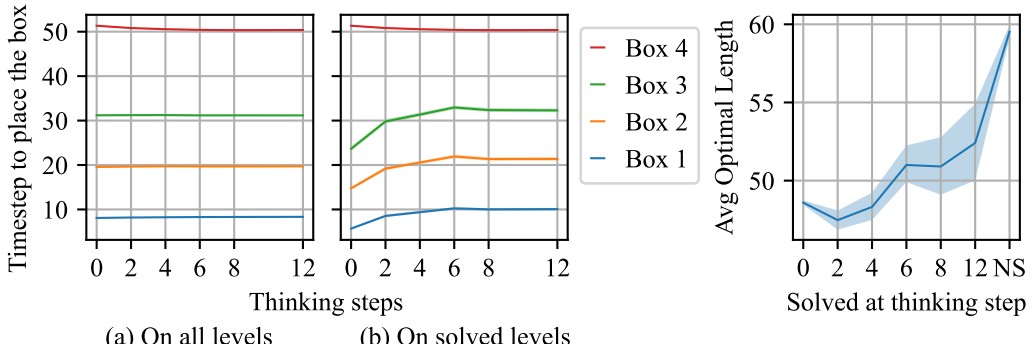

(a) On all levels  (b) On solved levels

Figure 5: **Left:** Average time step to place each box $B_i$ on target for different numbers of thinking steps. **(a)** Averages across all levels where the box $B_i$ is placed on target by DRC with $N$ thinking steps. **(b)** Averages for all levels solved by $6$ thinking steps but not solved by $0$ thinking steps. More thinking steps makes the DRC avoid greedy strategies in favor of long-term return. **Right:** Average optimal solution length of levels grouped by the number of thinking steps at which the level is first solved. Levels that take longer to solve tend to be harder. NS stands for "not solved".

Over the remaining training time, the RNN benefits more from thinking steps for the hard levels, but less for the medium levels; indicating perhaps that it learns better heuristics for them.

We also find that thinking time disproportionately helps with levels in which the agent can greedily put a box to target early, but where doing so is detrimental to solving the level. Figure 5 (left, b) shows this: with 0 thinking steps, the first box is pushed to target on average 4 steps too early; compared to the solution reached at 6 thinking steps. Some of this is driven by the forced thinking steps preventing the DRC$(3, 3)$ from taking a catastrophic action in the first few moves. However, the average time to first box is more than 6, even for 0 thinking steps, so there must also be many cases in which the 6 extra thinking steps give the NN time to find a better solution, rather than just preventing it from greedily pushing the first box. It must also be less greedy for boxes 2-3.

## 6.2 GENERALIZING BEYOND $10 \times 10$ INPUTS AND TRAINING EXAMPLES

This section goes beyond the capabilities of DRC$(3, 3)$ as a black-box unit. By looking for interpretable features in the activations of the ConvLSTM core, we found that layer 3, at the last tick, has four channels which represent the next action to take. Since the network up until the ConvLSTM

core is completely convolutional, we can evaluate it on inputs of *any* size – unlike DRC$(3, 3)$ as a whole, which only works on $10 \times 10$ inputs, and is trained only on levels with four boxes.

**Spatial aggregation.**    We start with the Layer 3 grid-wise binary next-action probes for each action from table 10. For each spatial location, these predict the action the DRC$(3, 3)$ will take in this step. We aggregate the spatial grid-wise predictions with three different methods: mean-pooling, max-pooling, and the proportion of locations with positive probe readouts for a particular action. We learn their relative weights and a bias for each action by optimizing the cross-entropy loss against the actions predicted by the MLP block. We use the Adam optimizer with learning rate $\ell = 10^{-3}$, annealed linearly to 0 for 10000 steps. Table 5 lists the learned parameters, which place most weight on the *mean-pooling* aggregation rule. The training set consists of 3000 levels, 1000 from each of the Boxoban unfiltered, medium, and hard training sets.

**Results.**    The aggregated features obtain $83\%$ and $77.9\%$ accuracy at predicting the MLP's action output on the training set and the medium-difficulty validation levels, respectively. This is less than expected given the $90\%+$ F1 scores of individual action probes in table 10.

The adjusted ConvLSTM is able to solve many levels that are out-of-distribution: larger than $10 \times 10$ in both dimensions, and with more than four boxes, e.g. those in fig. 2 (right) and fig. 8. Out of our test set collected from Þorsteinsson (2009), it can solve $59/480$ levels with both dimensions $> 10$, $30/203$ levels with only one dimension $> 10$ and $180/482$ levels with both dimensions $\leq 10$. Appendix B has a breakdown by level collection.

# 7    RELATED WORK

**Interpreting agents and planning.**    Several works have attempted to find the mechanism by which a simple neural network does planning in mazes (Mini et al., 2023; Knutson et al., 2024; Brinkmann et al., 2024), gridworlds (Bloom & Colognese, 2023), and graph search (Ivanitskiy et al., 2023). Men et al. (2024) investigates LLM reasoning in a simple block-stacking task. We believe the DRC we present is a clearer example of an agent than what these works focus on, and should be similarly possible to interpret.

Concurrent work (Anonymous, 2025) interprets another DRC Sokoban agent and can also predict future actions. That team invented grid-wise probes and the Agent-Directions and Box-Directions targets.

Other works interpret superhuman game-playing agents. Jenner et al. (2024) find evidence of lookahead in Leela Chess Zero (Team, 2018), without predicting future actions. McGrath et al. (2021); Schut et al. (2023) find interpretable concepts in AlphaZero (Silver et al., 2017).

**Probing for world models.**    Planning usually requires a model of the world. Li et al. (2023); Nanda et al. (2023b); Karvonen (2024) examine agents trained to predict possible moves in games, and are able to probe for the board state. Wijmans et al. (2023) also probe the state of navigation neural networks. Gurnee & Tegmark (2023) find a representation of positions in Earth for location-associated tokens in an LLM.

**Goal misgeneralization and mesa-optimizers for alignment.**    From the alignment perspective, AIs optimizing monomaniacally for a goal have long been a concern (e.g. Yudkowsky, 2006; Omohundro, 2008; see the preface of Russell, 2019). In a machine learning paradigm (Hubinger et al., 2019), the goal of the training system is not necessarily optimized; instead, the NN may optimize for a related or different goal (Di Langosco et al., 2022; Shah et al., 2022) or for no goal at all.

**Neural network architectures that reason.**    Many papers try to enhance NN thinking by altering the training setup or architecture. Schwarzschild et al. (2021b;a); Bansal et al. (2022) improve and evaluate a specific kind of RNN training that produces algorithmic thinking, though Knutson et al. (2024) argue that they do not generalize enough and thus do not implement the correct algorithm. Other works endow RNNs with variable amounts of computation per step, with (Chung et al., 2024) or without (Graves, 2016) an explicit world model.

**Systematic Generalization.**   Previous work has identified certain conditions like diverse datapoints and egocentric environments, under which neural networks generalize systematically (Lake & Baroni, 2023; Hill et al., 2020; Mutti et al., 2022). Similar interpretability work as ours can be done across many such neural networks to find common planning mechanisms and the conditions in which they emerge.

## 8   CONCLUSION

We replicated a small RL-based Sokoban agent, whose performance benefits from increased compute in the form of externally imposed no-op steps (Guez et al., 2019). We find that the agent learns to explicitly exploit this capability by pacing, which *substitutes* for the external no-op thinking steps. Extra computation helps with longer levels and those that require non-myopic thinking.

We show that the agent causally represents *plans* (sequences of actions to take). The most fundamental plan representation we find is the box-direction probes, which can predict and control where the agent pushes a box. By building up a chain of box probe steps over time, the agent is able to figure out how to place boxes to solve the level. Secondarily, we find agent-direction probes that take lower precedence than the box-direction probes, but can control the agent moves in isolated setups. We additionally find that the box probe chains get longer and longer with more thinking steps (fig. 3).

Finally, we find that extracting a decision from the plan takes place almost entirely within the ConvLSTM, with the MLP doing little more than decoding it. To demonstrate this, we show that the agent can generalize to levels bigger than what it had ever seen during training by manually replacing its MLP decoder based on our mechanistic understanding.

We hope the model organism we trained and open-source will catalyze research into NNs that learn to plan, and to understand their inner objectives.

### REPRODUCIBILITY STATEMENT

We provide the code and the trained models in the supplementary materials. We open-source our code and trained model(s). Additionally, sections 2, 4.1 and appendices A, E, and F describe the specific experimental details and settings we used to carry out our experiments.

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

## A    TRAINING THE TEST SUBJECT

**DRC**$(D, N)$ **architecture.** Guez et al. (2019) introduced the Deep Repeating ConvLSTM (DRC), whose core consists of $D$ convolutional LSTM layers with 32 channels and $3 \times 3$ filters, each applied $N$ times per time step. Our DRC$(3, 3)$ – or just DRC for brevity – has 1.29M parameters. Before the LSTM core, two convolutional layers (without nonlinearity) encode the observation with $4 \times 4$ filters.

The LSTM core uses $3 \times 3$ convolutional filters, and a nonstandard `tanh` on the output gate (Jozefowicz et al., 2015). Unlike the original ConvLSTM (Shi et al., 2015), the input to each layer of a DRC consists of several concatenated components:

- The encoded observation is fed into each layer.

- To allow spatial information to travel fast in the ConvLSTM layers, we apply *pool-and-inject* by max- and mean-pooling the previous step's hidden state. We linearly combine these values channel-wise before feeding them as input to the next step.

- To avoid convolution edge effects from disrupting the LSTM dynamics, we feed in a $12 \times 12$ channel with zeros on the inside and ones on the boundary. Unlike the other inputs, this one is not zero-padded, maintaining the output size.

**ResNet architecture.** This is a convolutional residual neural network, also from Guez et al. (2019). It serves as a non-recurrent baseline that can only think during the forward pass (no ability to think for extra steps) but is nevertheless good at the game. The ResNet consists of 9 blocks, each with $4 \times 4$ convolutional filters. The first two blocks have 32 channels, and the others have 64. Each block consists of a convolution, followed by two (relu, conv) sub-blocks, each of which splits off and is added back to the trunk. The ResNet has 3.07M parameters.

**Value and policy heads.** After the convolutions, an affine layer projects the flattened spatial output into 256 hidden units. We then apply a ReLU and two different affine layers: one for the actor (policy) and one for the critic (value function).

**RL training.** We train each network for 2.003 billion environment steps[5] using IMPALA (Espeholt et al., 2018; Huang et al., 2023). For each training iteration, we collect 20 transitions on 256 actors using the network parameters from the previous iteration, and simultaneously take a gradient step. We use a discount rate of $\gamma = 0.97$ and V-trace $\lambda = 0.5$. The value and entropy loss coefficients are 0.25 and 0.01. We use the Adam optimizer with a learning rate of $4 \cdot 10^{-4}$, which linearly anneals to $4 \cdot 10^{-6}$ at the end of training. We clip the gradient norm to $2.5 \cdot 10^{-4}$. Our hyperparameters are mostly the same as Guez et al. (2019); see appendix A.1.

**A\* solver.** We used the A\* search algorithm to obtain optimal solutions to each Sokoban puzzle. The heuristic was the sum of the Manhattan distances of each box to its nearest target. Solving a single level on one CPU takes anywhere from a few seconds to 15 minutes.[6]

## A.1 Training hyperparameters

All networks were trained with the same hyperparameters, which were tuned on a combination of the ResNet and the DRC$(3, 3)$. These are almost exactly the same as Guez et al. (2019), allowing for taking the *mean* of the per-step loss instead of the sum.

**Time limits.** During training, we want to prevent strong time correlations between the returns, so the gradient steps are not correlated over time. For this reason, the time limit for each episode is uniformly random between 91 and 120 time steps.

**Loss.** The value and entropy coefficients are 0.25 and 0.01 respectively. It is very important to *not* normalize the advantages for the policy gradient step.

**Gradient clipping and epsilon** The original IMPALA implementation, as well as Huang et al. (2023), *sum* the per-step losses. We instead average them for more predictability across batch sizes, so we had to scale down some parameters by a factor of $1/640$: Adam $\epsilon$, gradient norm for clipping, and L2 regularization).

---

[5]A rounding error caused this to exceed 2B (appendix A.2).

[6]The A\* solutions may be of independent interest, so we make them available at `https://huggingface.co/datasets/AlignmentResearch/boxoban-astar-solutions/`.

**Weight initialization.** We initialize the network with the Flax (Heek et al., 2023) default: normal weights truncated at 2 standard deviations and scaled to have standard deviation $\sqrt{1/\text{fan\_in}}$. Biases are initialized to 0. The forget gate of LSTMs has 1 added to it (Jozefowicz et al., 2015). We initialize the value and policy head weights with orthogonal vectors of norm 1. Surprisingly, this makes the variance of these unnormalized residual networks decently close to 1.

**Adam optimizer.** As our batch size is medium-sized, we pick $\beta_1 = 0.9$, $\beta_2 = 0.99$. The denominator epsilon is $\epsilon = 1.5625 \cdot 10^{-7}$. Learning rate anneals from $4 \cdot 10^{-4}$ at the beginning to $4 \cdot 10^{-6}$ at 2,002,944,000 steps.

**L2 regularization.** In the training loss, we regularize the policy logits with L2 regularization with coefficient $1.5625 \times 10^{-6}$. We regularize the actor and critic heads' weights with L2 at coefficient $1.5625 \times 10^{-8}$. We believe this has essentially no effect, but we left it in to more closely match Guez et al. (2019).

**Software.** We base our IMPALA implementation on Cleanba (Huang et al., 2023). We implemented Sokoban in C++ using Envpool (Weng et al., 2022) for faster training, based on gym-sokoban (Schrader, 2018).

## A.2 NUMBER OF TRAINING STEPS

In the body of the paper we state the networks train for 2.003B steps. The exact number is 2 002 944 000 steps. Our code and hyperparameters require that the number of environment steps be divisible by 5 120 = 256 environments × 20 steps collected, because that is the number of steps in one iteration of data collection.

However, 2B is divisible by 5 120, so there is no need to add a remainder. We noticed this mistake once the networks already have trained. It is not worth retraining the networks from scratch to fix this mistake.

At some point in development, we settled on 80 025 600 to approximate 80M while being divisible by 256 × 20 and 192 × 20. Perhaps due to error, this mutated into 1 001 472 000 as an approximation to 1B, which directly leads to the number we used.

## A.3 LEARNING CURVE COMPARISON

It is difficult to fully replicate the results by Guez et al. (2019). Chung et al. (2024) propose an improved method for RL in planning-heavy domains. They employ the IMPALA DRC$(3, 3)$ as a baseline and plot its performance in Chung et al. (2024, Figure 5). They plot two separate curves for DRC$(3, 3)$: that from Guez et al. (2019), and a decent replicated baseline. The baseline is considerably slower to learn and peaks at lower performance.

We did not innovate in RL, so were able to spend more time on the replication. We compare our replication to Guez et al. (2019) in appendix A.3, which shows that the learning curves for DRC$(3, 3)$ and ResNet are compatible, but not the one for DRC(1,1). Our implementation also appears much less stable, with large error bars and large oscillations over time. We leave addressing that to future work.

The success rate in appendix A.3 is computed over 1024 random levels, unlike the main body of the paper. Table 3 reports test and validation performance for the DRC and ResNet seeds which we picked for the paper body.

The parameter counts (table 2) are very different from what Guez et al. (2019) report. In private communication with the authors, we confirmed that our architecture has a comparable number of parameters, and some of the originally reported numbers are a typographical error.

## B GENERALIZING THE DRC$(3, 3)$ TO LARGER LEVELS

We license the levels by Þorsteinsson (2009) as GPLv3, and make them available in the supplementary material. The performance of the DRC$(3, 3)$ on each set of levels is in table 4. The sets with higher

Table 2: Parameter counts for each architecture.

| Architecture | Parameter count | |
| --- | --- | --- |
| DRC(3, 3) | 1,285,125 | (1.29M) |
| DRC(1, 1) | 987,525 | (0.99M) |
| ResNet: | 3,068,421 | (3.07M) |

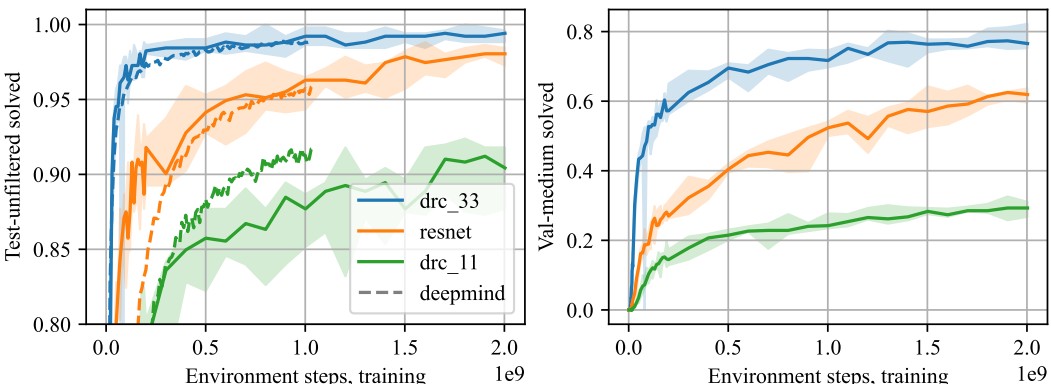

Figure 6: Success rate for Test-unfiltered and Validation-medium levels vs. environment steps of training. Each architecture has 5 random seeds, the solid line is the pointwise median and the shaded area spans from the minimum to the maximum. The dotted lines are data for the performance of architectures extracted from the (Guez et al., 2019) PDF file. The values are slightly different from what fig. 3 and section 6 report because they are calculated on a random sample of 1024 levels (24 levels are repeated for test-unfiltered, which only has 1000 levels).

Table 3: Success rate and return of DRC and ResNet on the unfiltered test set at various training environment steps.

| TRAINING ENV | TEST UNFILTERED | | | | VALID MEDIUM | | | |
| --- | --- | --- | --- | --- | --- | --- | --- | --- |
| STEPS | RESNET | | DRC(3, 3) | | RESNET | | DRC(3, 3) | |
| | SUCCESS | RETURN | SUCCESS | RETURN | SUCCESS | RETURN | SUCCESS | RETURN |
| 100M | 87.8 | 8.13 | 95.4 | 9.58 | 18.6 | -6.59 | 47.9 | -0.98 |
| 500M | 93.1 | 9.24 | 97.9 | 10.21 | 39.7 | -2.64 | 66.6 | 2.62 |
| 1B | 95.4 | 9.75 | 99.2 | 10.47 | 50.0 | -0.64 | 70.4 | 3.40 |
| 2B | 97.9 | 10.29 | 99.3 | 10.52 | 59.4 | 1.16 | 76.6 | 4.52 |

scores are also those which humans find easier: for example, "Dimitri & Yorick" was made for children by Jacques Duthen and consists entirely of relatively small levels (largest is $12 \times 10$) with at most 5 boxes. The sets where the DRC$(3, 3)$ solves nothing are also difficult for the authors of this paper.

We tried giving the DRC$(3, 3)$ from 2 to 128 extra thinking steps for these levels, incrementing in powers of two. For most sets, we find some benefit to 2-4 thinking steps, but no more. The sole exception is XSokoban, which contains one level (fig. 8(c)) which requires 128 steps of thinking to solve.

We encourage the reader to go to the website by Þorsteinsson (2009) and try solving the levels.

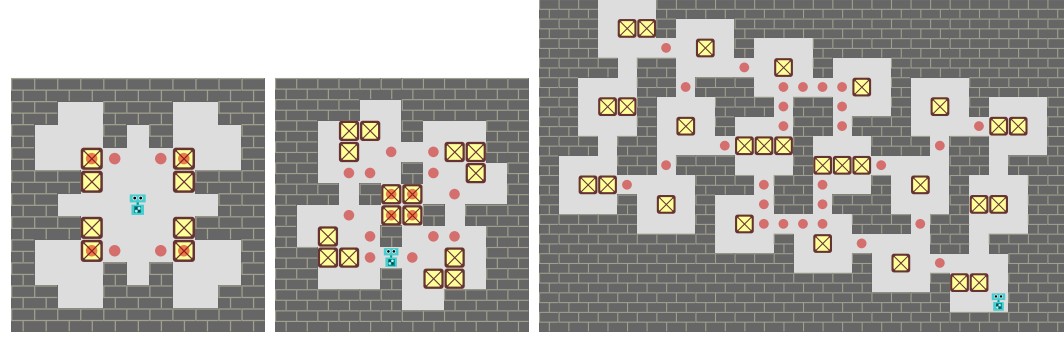

(a) Microban, level 105      (b) Microban, level 144      (c) Mas Sasquatch, level 15

Figure 7: Thinking steps are not always useful: For level (a), the network succeeds with 0 to 16 thinking steps, but fails with 32 or more thinking steps. Levels (b) and (c) are too difficult for the DRC$(3, 3)$, and it always fails on them independent of thinking steps, though it comes up with decent partial solutions. All levels from Þorsteinsson (2009).

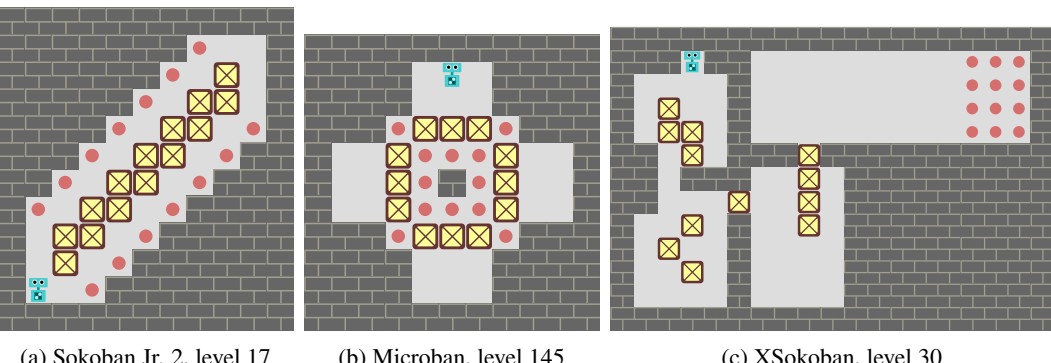

(a) Sokoban Jr. 2, level 17      (b) Microban, level 145      (c) XSokoban, level 30

Figure 8: Some levels require thinking steps for success: For level (a), the network always succeeds. For level (b), it succeeds at 32 or more thinking steps. For level (c), it needs 128 thinking steps to succeed. All levels from Þorsteinsson (2009).

## C  BISTABLE AND UNSTABLE PLANS IN TOY ENVIRONMENTS

The causal intervention results from table 1 show that the box-directions probe is much more causal than the agent-directions probe. We further study this in an isolated setting. Figure 9 (left) shows a level with only a single box next to a target, where the agent has only two equally good paths it can follow. Without intervening, the agent first takes the path on the right by moving three steps, while the agent-direction probe grows towards the box. After the first three moves, the chain connects. The agent then takes a U-turn to follow the path on left. This is very unlike the optimal behavior, which is to commit to any path and follow it.

Table 4: Performance of the DRC$(3, 3)$ on each set of levels by (Þorsteinsson, 2009). The "Max solved" columns represent the proportion of levels solved at the number of steps in the "at steps" column, which is the highest solved proportion for each number of thinking steps tried.

| LEVEL COLLECTION | # | ALL LEVELS | | | # | LEVELS LARGER THAN $10 \times 10$ | | |
|---|---|---|---|---|---|---|---|---|
| | | SOLVED | MAX SLV. | MAX AT | | SOLVED | MAX SLV. | MAX AT |
| Dimitri & Yorick | 61 | 83.6% | 88.5% | 16 | 0 | — | — | — |
| Sokoban Jr. 1 | 60 | 80.0% | 81.7% | 2 | 19 | 73.7% | 73.7% | 0 |
| Howard's 3rd set | 40 | 60.0% | 62.5% | 4 | 1 | 0.0% | 0.0% | 0 |
| Simple sokoban | 61 | 54.1% | 60.7% | 16 | 51 | 47.1% | 54.9% | 16 |
| Sokoban Jr. 2 | 54 | 48.1% | 50.0% | 2 | 40 | 45.0% | 47.5% | 2 |
| Microban | 155 | 23.9% | 24.5% | 4 | 17 | 5.9% | 11.8% | 2 |
| Deluxe | 55 | 21.8% | 23.6% | 2 | 1 | 0.0% | 0.0% | 0 |
| Sokogen 990602 | 78 | 20.5% | 23.1% | 8 | 0 | — | — | — |
| Yoshio Automatic | 52 | 13.5% | 17.3% | 2 | 0 | — | — | — |
| Sasquatch III | 16 | 6.2% | 6.2% | 0 | 8 | 0.0% | 0.0% | 0 |
| Howard's 1st set | 100 | 5.0% | 6.0% | 2 | 54 | 0.0% | 0.0% | 0 |
| Howard's 2nd set | 40 | 5.0% | 10.0% | 4 | 22 | 0.0% | 0.0% | 0 |
| Microcosmos | 40 | 5.0% | 7.5% | 4 | 0 | — | — | — |
| Still more levels | 35 | 2.9% | 2.9% | 0 | 34 | 2.9% | 2.9% | 0 |
| Sasquatch IV | 36 | 2.8% | 2.8% | 0 | 20 | 0.0% | 0.0% | 0 |
| Xsokoban | 40 | 2.5% | 5.0% | 128 | 39 | 2.6% | 5.1% | 128 |
| Sasquatch | 49 | 2.0% | 4.1% | 4 | 39 | 0.0% | 2.6% | 4 |
| David Holland 1 | 10 | 0.0% | 0.0% | 0 | 5 | 0.0% | 0.0% | 0 |
| David Holland 2 | 10 | 0.0% | 0.0% | 0 | 9 | 0.0% | 0.0% | 0 |
| Howard's 4th set | 32 | 0.0% | 0.0% | 0 | 30 | 0.0% | 0.0% | 0 |
| Mas Sasquatch | 50 | 0.0% | 0.0% | 0 | 43 | 0.0% | 0.0% | 0 |
| Nabokosmos | 40 | 0.0% | 0.0% | 0 | 0 | — | — | — |
| Sokoban | 50 | 0.0% | 0.0% | 0 | 48 | 0.0% | 0.0% | 0 |

We can get the agent to commit to one path early by intervening with the agent-direction probe on any of the two paths *on the first step only*, as shown in fig. 9 (middle). When we construct a similar level where there are two equally good paths for a box to follow, we find that the agent-direction probe is no longer causal; but the boxes direction probe *does* get the agent to commit. Figure 9 (right) shows an almost empty level. When we intervene with the box-directions probe on the first four steps, the agent follows the laid out path in those steps, but then quickly switches to a more optimal path with fewer turns. The laid out path (not taken, shown in red) is not followed after stopping the intervention.

This analysis provides strong evidence that the network primarily builds upon the boxes directions represented in the activations and relies on the agent-direction when the boxes directions do not inform the subsequent actions.

# D  CASE STUDIES

**Case Study: Thinking makes some levels solvable fig. 10(a).**  For example, thinking lets the DRC solve fig. 10(a). In the no-thinking condition, the DRC first pushes box $C$ one square to the right. It then goes back to push $A$ to $a$, but it has now become impossible to push box $B$ onto $b$. In contrast, after thinking, the DRC pushes $A$ to $a$ first, which lets it solve the level.

**Case Study: Thinking speeds up solving fig. 10(b).**  fig. 10(b) illustrates a scenario where the agent happens. In the no-thinking condition, the DRC takes many steps back and forth before pushing any boxes. First it goes down to $y$, up to $c$, then down onto $z$, back up to $y$ and to $z$ again. It then proceeds to solve the rest of the puzzle: push box $A$ onto $a$, prepare box $B$ on $x$ and box $C$ where $B$ originally was, push in boxes $B, C$ and finally $D$. In the thinking condition, the DRC makes a beeline for $A$ and then plays the same solution.

**Case Study: Thinking slows downfig. 10(c).**  Figure 10(c) illustrates a scenario where thinking time results in a slower solution. In the no thinking condition, the DRC starts by pushing box C into

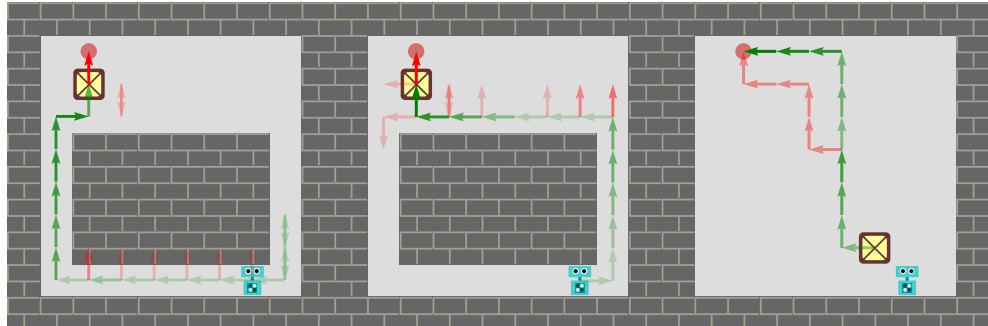

Figure 9: **Left:** A custom level where the agent has two equally good paths to follow. The arrows show the prediction of the agent-direction probe with opacity proportional to the number of times an arrow was predicted across all the steps. The green and red arrows are correct and incorrect predictions, respectively. The agent behaves suboptimally by returning to the start and going left after first going right for three steps. **Middle:** The agent takes the path on the right after intervening with the corresponding arrows using the agent-direction probe on the first step. The same happens on the left if we intervene on that path without the suboptimal steps right of the undisturbed agent. **Right:** An empty level with box-directions probe intervening on the first four steps which are followed correctly by the agent on those four steps. When the intervention is removed on the fifth step, the agent computes the simpler path in green and doesn't follow the path laid out earlier (in red).

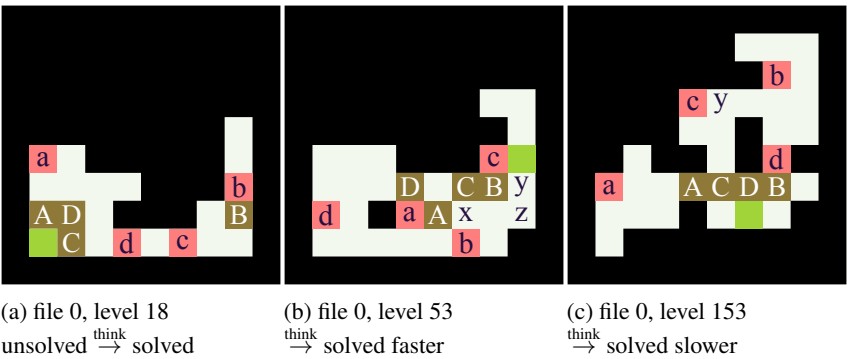

(a) file 0, level 18
unsolved $\overset{\text{think}}{\to}$ solved

(b) file 0, level 53
$\overset{\text{think}}{\to}$ solved faster

(c) file 0, level 153
$\overset{\text{think}}{\to}$ solved slower

Figure 10: Case studies of three medium-validation levels demonstrating different behaviors after 6 thinking steps. Colors are as in fig. 12. Boxes and targets are paired in upper- and lower-case letters respectively, such that the DRC's best solution places boxes on targets in alphabetical order. Videos available at this https URL. Levels solved faster obtain higher return as the agent incurs the per-step penalty fewer times. Note that the letters used are for reference and not a property of Sokoban.

position y, then pushes boxes A, then B into place. On the way back down, the DRC pushes C onto c and finally D onto d. In contrast, in the thinking condition, the DRC goes the other way and starts by pushing B onto b. The subsequent solution (A, C, then D) is the same, but the DRC has wasted time trekking back from B to A.

# E   PROBE TRAINING

**Probe architecture**   We train multiple different linear probes on the hidden states $h$ and cell states $c$ activations of the network. The probes are trained either on the states of the individual layers or by concatenating the states of all the three layers. We collect the activations of the model by letting it play through the hard levels. The training set for the probes are constructed by taking a random sample of cached levels and including all the timesteps in the level excluding the first five timesteps. We exclude these intial steps as we believe they can be noisy as the network comes up with the stategy to solve the levels.

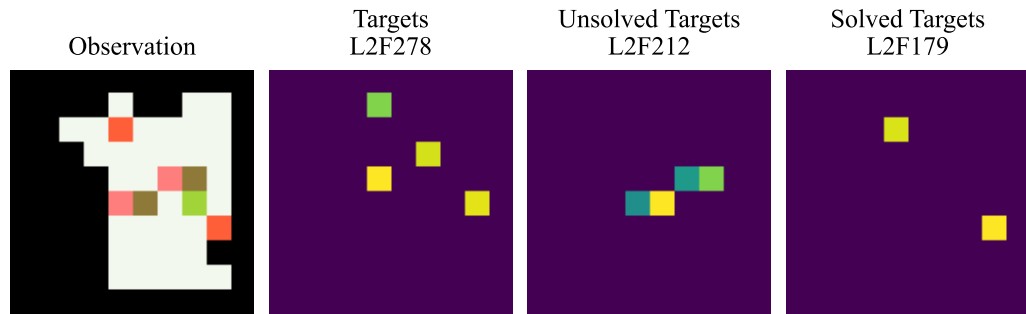

Figure 11: Visualization of some interpretable features from the SAE of last layer. These features also appear monosemantically in the channels. The precision, recall, and F1 score for the features are reported in table 9.

The probes are trained with logistic regression with L1 decay using the Scikit-Learn library. We do a grid-search on the learning rate and the L1 weight decay to select the probe that has the highest F1-score on a validation set.

For multi-class probe targets, each potential output class $l$ is treated like a different data point. That is: for each label $l$, we consider a prediction positive if the highest probe logit is for label $l$, and negative otherwise. We consider a data point positive iff the true label is label $l$. Then, we compute the confusion matrix and the F1 score from the $n \cdot l$ data points.

Table 5: Weight and bias for transforming direction probes to predictions

|  | Mean | Max | Positive proportion |
| --- | --- | --- | --- |
| Weight | 1.2086 | -0.0582 | 0.2070 |

|  | Up | Down | Left | Right |
| --- | --- | --- | --- | --- |
| Bias | 0.3337 | -0.0921 | -0.0632 | -0.0539 |

## F  LOOKING FOR INTERPRETABLE FEATURES WITH SPARSE AUTOENCODERS (SAEs)

In order to search for more monosemantic and interpretable features in the network, we train sparse-autoencoders (SAEs) (Huben et al., 2023; Bricken et al., 2023) on the individual squares in the $h$ hidden state of the network consisting of 32 neurons. Thus, we get a $10 \times 10$ visualization for each SAE feature as shown in Figure 11. We use the top-k activation function (Gao et al., 2024; Tamkin et al., 2024) for the SAE as it is currently the state-of-the-art method to train SAEs and is easier to work with as it directly sets the $L_0$ norm of the SAE activations. The hyperparameter search space for training the SAE is provided in Table 6. We train separate SAEs for each layer with the specified hyperparameters and pick the one that achieves greater than $90\%$ explained variance while having interpretable features assessed through manual visual inspection. We release these trained SAEs and probes in the same huggingface repo as our trained DRC networks. [7]

We provide examples of some of the interpretable features from an SAE trained on the last layer with $k = 8$ in table 7 with corresponding visualization in fig. 11. For the "Target", "Unsolved", and "Solved" concepts (table 9), we observed cases where the SAE feature is offset from the ground truth by 1 square in either horizontal or vertical directions. The level setup includes a permanent one-square outer-edge wall, so this offset never results in an out-of-bounds issue. We evaluated these potential "Offset" variants for the Target, Unsolved, and Solved concepts.

---

[7] https://huggingface.co/AlignmentResearch/learned-planner

All of the features in the SAE that we find to be interpretable are already embedded in individual channels. These channels are either as monosemantic as the SAE features, or in some cases more monosemantic as measured by the F1 score. Table 10 reports the precision, recall, and F1 scores for the action features as evaluated through channels, SAE neurons, and linear probe trained against the ground truth action predictions. Table 9 reports the same scores for additional interpretable features. The SAE action features have lower F1 scores than channels by a margin of $5.9\%$ on average. On the other hand, the linear probes trained across all channels of the hidden state have similar F1 scores as the specified channels indicating that the channels are monosemantic and cannot be improved upon by combining various channels using a linear probe.

Table 6: Hyperparameter search space for training SAE

| HYPERPARAMETER | SEARCH SPACE |
|---|---|
| $k$ | $\{4, 8, 12, 16\}$ |
| learning rate | $\{1e-5, 5e-5, 1e-4, 5e-4, 1e-3\}$ |
| expansion factor | $\{16, 32, 64\}$ |

Table 7: SAE Feature Concepts

| CONCEPT | DESCRIPTION |
|---|---|
| Target | The 4 target squares (static) |
| Unsolved | Targets and boxes that aren't solved |
| Solved | Solved target squares with a box on them |
| Agent Up | The agent will move Up next step |
| Agent Down | The agent will move Down next step |
| Agent Left | The agent will move Left next step |
| Agent Right | The agent will move Right next step |

Table 8: Breakdown of levels by category at 6 thinking steps.

| LEVEL CATEGORIZATION | PERCENTAGE |
|---|---|
| Solved, previously unsolved | 6.87 |
| Unsolved, previously solved | 2.23 |
| Solved, with better returns | 18.98 |
| Solved, with the same returns | 50.16 |
| Solved, with worse returns | 5.26 |
| Unsolved, with same or better returns | 15.14 |
| Unsolved, with worse returns | 1.36 |

## G    ADDITIONAL QUANTITATIVE BEHAVIOR FIGURES AND TABLES

## H    ADDITIONAL RELATED WORK

**Ethical treatment of AIs.**    Do AIs deserve moral consideration? Schwitzgebel & Garza (2015) argue that very human-like AIs are conceivable and clearly deserve rights. Tomasik (2015) suggests that most AIs deserve at least a little consideration, like biological organisms of any species (Singer, 2004). But what does it mean to treat an AI *ethically*? Daswani & Leike (2015) argue that the way to measure pleasure and pain in a reinforcement learner is not by its absolute amount of return, but rather by the temporal difference (TD) error: the difference between its expectations and the actual return it obtained. If the internals of the NN have a potentially different objective (Hubinger et al., 2019; Di Langosco et al., 2022), then the TD error should come from a place *other* than the critic head. *Research into learned search algorithms* is an early step toward finding the learned-reward

Table 9: Scores for SAE and Channel features

| CONCEPT | OFFSET $(dy, dx)$ | CHANNEL | | | | SAE FEATURE | | | |
|---------|------------------|---------|------|------|------|-------------|------|------|------|
| | | NUMBER | PREC | REC | F1 | NUMBER | PREC | REC | F1 |
| Target | (1, 0) | L3C17 | 97.8 | 97.7 | 97.8 | L3F278 | 97.8 | 98.1 | **98.0** |
| Unsolved targets and boxes | (0, 0) | L3C7 | 94.9 | 90.8 | **92.8** | L3F212 | 95.3 | 86.6 | 90.7 |
| Solved targets | (0, 0) | -L3C7 | 91.6 | 94.6 | **93.0** | L3F179 | 91.7 | 91.5 | 91.6 |

Table 10: Action features scores across channels, probes, and SAE features

| FEATURE | CHANNEL | | | | SAE FEATURE | | | | PROBE | | |
|---------|---------|------|------|------|-------------|------|------|------|------|------|------|
| | NUMBER | PREC | REC | F1 | NUMBER | PREC | REC | F1 | PREC | REC | F1 |
| Up | L3C29 | 95.7 | 88.1 | **91.7** | L3F270 | 93.9 | 76.2 | 84.1 | 97.5 | 86.5 | **91.7** |
| Down | L3C8 | 98.4 | 80.8 | 88.8 | L3F187 | 98.0 | 79.1 | 87.6 | 97.6 | 86.9 | **91.9** |
| Left | L3C27 | 85.5 | 84.6 | **85.1** | L3F244 | 96.1 | 63.2 | 76.2 | 83.5 | 86.6 | 85.0 |
| Right | L3C3 | 97.0 | 86.9 | 91.7 | L3F385 | 94.6 | 78.5 | 85.8 | 97.6 | 87.4 | **92.2** |

internal TD error, if it exists. This could be a higher-assurance complement to simply asking the AI (Perez & Long, 2023).

**Chain-of-thought faithfulness.** Large language models use chain of thought, but are they faithful to it, or do they think about their future actions in other ways (Lanham et al., 2023; Pfau et al., 2024)? One could hope that LLMs perform all long-term reasoning in plain English, allowing unintended human consequences to be easily monitored, as in Scheurer et al. (2023).

**Fully reverse engineering small networks.** Many works reverse engineer all of, or most of, a small NN that does an algorithmic task (Nanda et al., 2023a; Chughtai et al., 2023; Zhong et al., 2023; Quirke & Barez, 2023).

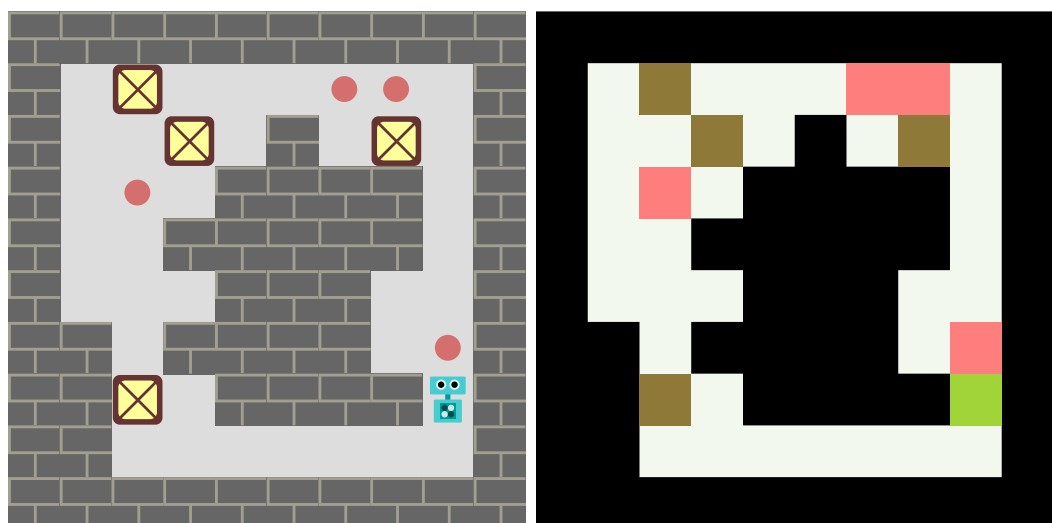

Figure 12: **Left:** A Sokoban level from the hard set. **Right:** the same level as the NN sees it, one pixel per tile. Walls are black, boxes are brown, targets are pink and the robot is green.

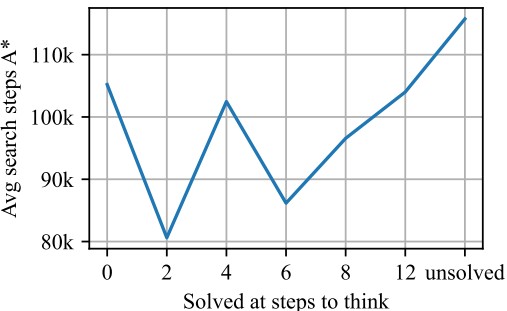

Figure 13: Number of thinking steps required to solve the level vs. number of nodes A* needs to expand to solve it. The trend is increasing at the end but very unclear, indicating different heuristics used by the DRC and A*.

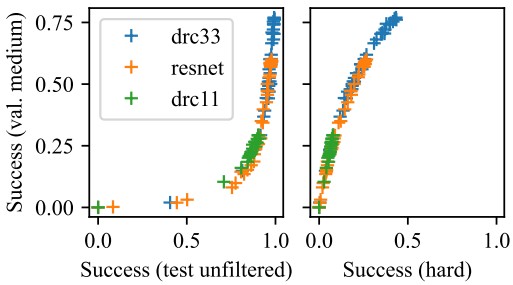

Figure 14: Success rate on datasets of various difficulty, for various checkpoints of each architecture. This deviates very little from a curve, which shows that ResNets and DRCs which are equally good at the easier sets are also equally good at the harder sets. Perhaps DRC(1,1) is a slight exception, but it also performs much worse than the others overall (see appendix A.3).

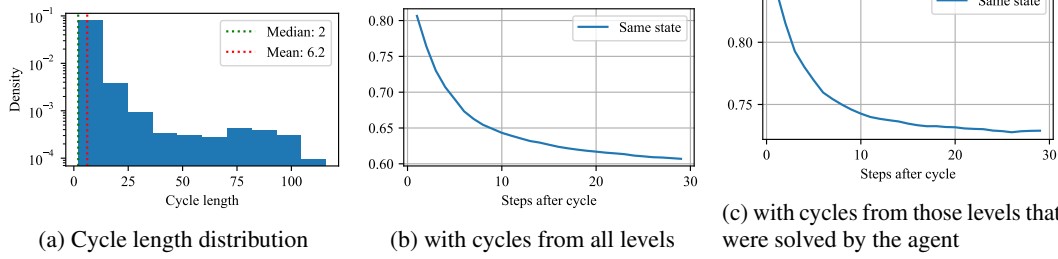

(a) Cycle length distribution    (b) with cycles from all levels    (c) with cycles from those levels that were solved by the agent

Figure 15: We replace $N$-length cycles with $N$ thinking steps and check for the same state after some timesteps. *(a)* A histogram of cycle lengths in the medium-validation set. *(b, c)* After replacing a cycle with the same length in thinking steps, are all the states the same for the next $x$ steps?

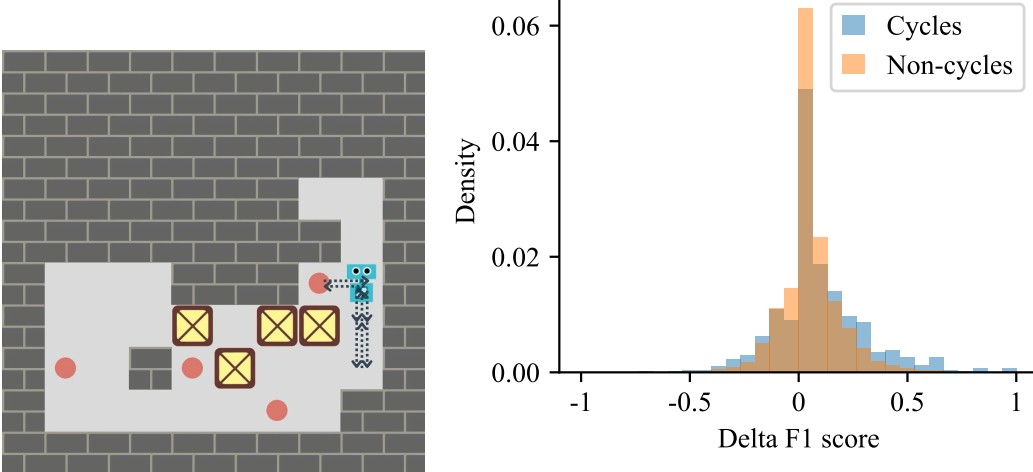

(a) Pacing behavior on file 0, level 53. On the given starting observation, the agent paces around 4 spaces in the first 9 steps and then goes on to solve the level. Video for the level is available at this url (removed for double-blind review).

(b) Change in per-step F1 score of box-directions probe for moves in cycles and outside cycles on medium-difficulty validation levels. The non-cycle moves were recorded from the same distribution of timesteps where cycles occur but from levels without a cycle at those steps. Mean per-step change in F1 for cycle and non-cycle steps are $1.40\% \pm 0.06\%$ and $0.84\% \pm 0.04\%$ respectively.

Figure 16: Illustration of cycles and F1 scores

