# OpenReview forum: "Planning in a recurrent neural network that plays Sokoban"
_ICLR.cc/2025/Conference — Submitted to ICLR 2025_

### Official Review · Reviewer_cMKh · 2024-11-01

**Soundness:** 1
**Presentation:** 2
**Contribution:** 2
**Rating:** 3
**Confidence:** 4

**Summary:**

The paper investigates an RL agent that can pace itself by taking extra computational steps in complex scenarios. The paper claims that by adjusting the network's hidden states, they extended its capabilities, allowing it to solve much larger and more challenging levels, and have open-sourced the compact, 1.29M-parameter model for further study in learned planning.

**Strengths:**

The claims of the paper are ambitious.

**Weaknesses:**

Some claims seem unrealistic, for instance in Figure 2, it is written "Left: XSokoban-31, which the DRC(3, 3) solves with 0 thinking steps", I'm not convinced that this claim is correct. Are the trained models already shared such that this could be checked? (beside the promise "We will open-source our code and trained model(s)").

Some words do not seem to have a clear meaning, e.g. in the abstract "perform model surgery" and "makes it an excellent model organism".

Some other parts are unclear:
- For instance, at the end of the introduction, it is written "The evidence highly suggests that the RNN implements an online search algorithm". What evidence exactly is there for such a "high suggestion"?
- Beginning of Section 3.1 "the extra thinking time disproportionately helps with more difficult levels": what does disproportionately mean in this case?

**Questions:**

See questions in the weaknesses about the unclear points of the paper.

---

> ### Author Response · Authors · 2024-11-20
>
> We are delighted that you found our work ambitious, novel and surprising. We are happy to assist you in replicating any of our claims.
>
> > Some claims seem unrealistic, for instance in Figure 2, it is written "Left: XSokoban-31, which the DRC(3, 3) solves with 0 thinking steps", I'm not convinced that this claim is correct. Are the trained models already shared such that this could be checked? (beside the promise "We will open-source our code and trained model(s)").
>
> Yes, we submitted one trained model (the one we examine) along with code as supplementary material to OpenReview.  We should have stated that instead of “will open-source”, apologies. For your ease of verification, we have updated the code and model weights format so it only requires Pytorch (not Jax) and contains all the dependencies, anonymized and in a runnable format.
>
> **Instructions for running the NN that solves XSokoban-31:** Get a fresh Python 3.11 environment and go to the uncompressed `supplementary-material` folder. Only CPU is needed. Then install everything:
>
> pip install -e MambaLens -e farconf -e gym-sokoban -e stable-baselines3 -e 'learned-planners[torch,dev-local]'
>
> And run the script: python learned-planners/plot/play_bigger_levels.py . This should run in under a minute and will create a `level_22_031.mp4` , where you can watch the DRC(3, 3) solve the X-sokoban-31 level. You can change the level/s to be generated by passing options to the script (run python learned-planners/plot/play_bigger_levels.py --help). We encourage you to read the code to check it does what we wrote in the paper.
>
> Please do not hesitate to write back if you have any problems running the script. We have tested on a fresh virtualenv in a Mac, and also inside the python:3.11 Docker container ( use `docker run -v $(pwd):/workspace -it python:3.11 /bin/bash` to get an environment and then run the commands above ).
>
>
> We originally also shared pickled probes and sparse autoencoder weights (SAEs) under the `learned-planners` subdirectory of the submitted supplementary material. You can also watch the generated videos, e.g. at `videos/bigger-levels/xsokoban-31-0.mp4` . We have also added the anonymised training code `train-learned-planners`, but we have not tested running it so it is mainly for you to read.
>
> We can answer any further questions you have and can assist you in reproducing the claims.
>
> It is worth noting that the levels depicted in Figure 2 are cherry-picked from the set of levels intended for humans. We have also cherry-picked examples of levels the DRC(3, 3) **does not** solve or for which thinking steps hurt in Figure 8, as well as a (non-cherry-picked) full breakdown in Table 6. As Table 6 indicates, the DRC(3, 3) can only solve these 2 levels (5.0% of 40) from XSokoban at its best, though there are many large levels from the other sets that it solves. On reflection we did not emphasize this enough, and will change “more examples in appendix B” to “For many example levels which the DRC does not solve, as well as a breakdown of solve rate by level collection, see appendix B”.
>
> > Some words do not seem to have a clear meaning, e.g. in the abstract "perform model surgery" and "makes it an excellent model organism".
>
> We borrow the terms from the ML interpretability community. Model organism originally comes from biology (https://en.wikipedia.org/wiki/Model_organism) and is used prominently in e.g. [Hubinger et al. (2024)](https://arxiv.org/pdf/2401.05566). In biology (interpretability), a model organism refers to a simpler organism (neural network) that can be studied with the expectation that discoveries made in the model organism will provide insight into the workings of other more complex organisms (neural networks).
>
> Model surgery is also employed in some interpretability works, e.g. [Wang et al. 2024](https://arxiv.org/abs/2407.08770) and [Li et al. 2023](https://arxiv.org/abs/2304.05653). It refers to changing the capabilities of the model in a targeted way, typically one that requires some knowledge about the algorithm it implements. It is possible we should call this ‘model editing’ (doesn’t have to be targeted) or ‘model stitching’. In any case we will refer to the earlier works. Thank you for pointing out that this is not a common term.

---

> > ### Author Response · Authors · 2024-11-20
> >
> > > "The evidence highly suggests that the RNN implements an online search algorithm". What evidence exactly is there for such a "high suggestion"?
> >
> > Most of the paper elaborates in this point, but here is the structure of the argument, summarized:
> > Section 4 shows the NN has a plan (we can read it out and write to it, and correspondingly predict or influence the NN’s future actions).
> > Fig. 5 (middle, right) show that this plan improves over time. Fig. 5 (left) shows that more test-time computation improves performance. So whatever thing the NN is doing, it is something that improves plans over time with more computation.
> > Section 3: We have OK evidence that the NN pacing around is important to its performance: we can substitute it for extra steps-to-think without actions, and training did not stamp it out even though it is bad for return. This adds up to a little boost to evidence for “extra compute improves performance”.
> > In the causal experiments, if we stop applying an arbitrary plan to the activations, the NN comes up with a better plan and follows that (Fig 6, right).
> > These are all characteristics that an online search algorithm would have. *In Bayesian terms, we can confidently conclude that the likelihood p(observations | the NN internally is doing online search) is pretty high.*
> >
> > However, that does not come close to proving that the NN internally is just doing online search. Perhaps some combination of heuristics exists that yields the same results on these tests, or the NN is doing search only at the beginning and then executing and we’re not overwriting its “cache of searched paths”, or other hypotheses we have not thought of. So the likelihood ratio p(observations | online search) / p(observations | everything else) is larger than 1 (we’d say its value is roughly 3-6) but not large enough to overwhelm even a 50:50 prior. Which is why we’re calling it a *high suggestion*.
> >
> > We’re interested in better characterizing the actual strength of the evidence, and in discussing the point of whether we should summarize this as “highly suggests”. In any case, we think the evidence is strong enough to be interesting.
> >
> > It’s also possible that we should claim the evidence highly suggests that the NN implements a *search* algorithm, without it necessarily being online.
> >
> > > Beginning of Section 3.1 "the extra thinking time disproportionately helps with more difficult levels": what does disproportionately mean in this case?
> >
> > Thank you for this question. It means that thinking time helps solve levels overall by some amount X, but helps solve harder levels some amount Y and Y>X.
> >
> > On the hard difficulty set, the network solves extra 7.1% levels with thinking steps, but only 4.7% extra levels for medium difficulty validation (see Figure 1 Bottom left).
> >
> > Figure 3 (right) also shows increasing thinking steps helps in solving levels with longer optimal solutions, which is a good proxy for them being harder levels. We will clarify this further on line 157 in the updated paper.

---

> > > ### Comment · Reviewer_cMKh · 2024-11-26
> > > **Thank you for the clarifications**
> > >
> > > I appreciate the effort of the authors to provide clarifications.
> > >
> > > I think most elements from the rebuttal makes sense. However, it would be useful if you make the changes in the paper. A new version of the paper could fix some of the elements such as the "high suggestion" that it performs online search" (as well as the comments from the other reviewers). In that case, it would increase the possibility that we change our scores.

---

### Official Review · Reviewer_p3dc · 2024-11-03

**Soundness:** 2
**Presentation:** 1
**Contribution:** 2
**Rating:** 3
**Confidence:** 3

**Summary:**

The paper studies the behaviour of a Deep Repeated ConvLSTM (DRC) network architecture that learns to play Sokoban. The paper specifically documents the phenomenon of “pacing” whereby the network moves in cycles before performing goal-directed actions. The authors interpret these behaviours as an indication that the model gives itself additional computational steps before the execution of a strategy. The authors also engage in an interpretability analysis of the models via the training of linear probes on LSTM states. The authors find that probes (logistic regressions) trained on these states are able to decode a variety of task relevant variables such as the future movements of boxes and agent movements for locations in the grid world. The performance of these probes varies over the different probe targets but are generally predictive of future actions. One specific type of probe, encoding the direction of boxes, is also successfully leveraged to intervene in the model activations to change (steer) model behaviour. Finally, the authors modify the DRC decoder to allow the model to somewhat successfully generalise to new puzzles of differing sizes and with variable number of boxes.

### Recommendation

I think the paper should be rejected because (1) the claims are not well supported, (2) it is unclear and unpolished in many places, and (3) I think quite disconnected from the literature it claims to be connected to. See more detailed criticisms above.

**Strengths:**

- **Nice task.** I think the use of Sokoban is interesting as it is a hard planning problem that has many failure modes for greedy models which perform actions without careful regard and deliberation of future actions. I think research that interprets how models perform planning is especially relevant now with current developments happening in the reasoning abilities of large (language) models.
- **A variety of interpretability techniques.** The authors make use of a variety of techniques of mechanistic interpretability.
- **Planning in a model free setting.** I think it is interesting that the authors attempt to explicitly study the ability to plan in a model-free agent, where historically planning has been in the camp of model-based RL. DRC represents a very flexible model class with minimal inductive biases. This flexibility leaves open if the model learns to plan in the traditional sense. It is an interesting question if model representations and computation might represent plans explicitly.

**Weaknesses:**

Despite these strengths I think the paper has many weaknesses and does not do a great job of elucidating if the model is planning. In addition, I think the paper is very unclear in many places.

- **Framing of results.** The paper is abundant with sections in which the authors make quite substantial and large claims. I.e. in the introduction claims that “linear probes show that activations represent a plan that causes actions” and that “the behaviour of the RNN suggests that it performs search”. However, Section 4 openly admits that they find little evidence to suggest that the model is performing search. I think the claims that the paper makes and the presented evidence are quite disconnected and need substantial revamping. I also think that a contribution statement would make clear what really is being contributed.
- **Strength of evidence.** I do not think some of the author's evidence is strong enough to support their claims. In Section 4 on linear probing the predictive power of the next box and next target probes are much lower than direction probes. I think intuitively should these not be a much better measure of planning depth as they are more relevant to long term plans? The results on generalization also appear somewhat limited as the performance is not as good as the intro and abstract might make you believe.
- **Novelty.** Section 5 shows some generalisation ability of DRCs but this has already been shown in a variety of settings by Guez et al. (2019) and I think it is not super clear what the current paper adds.
- **Clarity.** I think there are also places in which the elaboration of the methods used are not clear. For example, I found myself quite confused about the methods used in Section 5.1 *Spatial aggregation*. I was looking in the appendix but could not find an exact explanation of what was done here. What exactly is being aggregated? Also, what is the number column in Table 3?
- **Interpretability results left uninterpreted.** There are several places where the authors fail to give sufficient explanations, interpretation, or context for their findings. For example, the section on linear probes finds good predictiveness but *much* worse causal probe results. Why is this the case? I felt like I was just left wondering as the reader.  I am also wondering what is special about pacing as opposed to other forms of taking inconsequential actions that are not cyclical, both should yield the same result, right? I think the authors would have to give an explanation about what is special about pacing.
- **Unpolishedness.** There are several places where I felt that the paper was not well polished and presented. For example the caption in Figure 5 does not have labels for the subplots (i.e. left, right, centre) even though these are referred to in the main text. Similarly, in Figure 3 the x-axis labels are below the subplot caption, which makes the figure unclear. Further, often the terms NN, RNN, and DRC are used interchangeably. I think it would be good to just stick to one term referring to the model (DRC is used the majority of times). There is also the sentence “we did not bother collecting an i.i.d. test set” in Section 5, I personally think such language is not perfectly appropriate for a paper.
- **Placing in the literature.** I think the paper does quite a poor job of placing the paper in the literature. Especially the related work section (Section 6\) I find quite strange. For example it is not clear to me how the presented work connects to the “Ethical treatment of AIs”  or to “Goal misgeneralization and mesa-optimizers for alignment”.

### Potential Improvements
1. The claims should be appropriately fitted to the findings.
2. The clarity of the paper should be improved with respect to the methods used in Section 5.1.
3. The related works section should be more carefully written.
4. Perhaps reconsider having the section on sparse autoencoders in the main text. If they do not perform well then why have them with your main results as opposed to the appendix?

**Questions:**

1) Why does the causal probe perform so poorly compared to their predictive power?
2) What is special about pacing as opposed to other forms of moving around?
3) What is being aggregated over in Section 5.1.?

---

> ### Author Response · Authors · 2024-11-29
>
> Thank you for taking the time to review our paper.
>
> **Framing of results**. The specific claim we make in this paper is that we have strong evidence that the model has a *plan*, and weak evidence that it’s doing *search*. This matches the sentences you highlight: the ability to read/write the plan with probes is strong evidence that there is a plan, and behavioral evidence is weak.
>
> We agree that a clearer contribution statement was necessary. We have updated the paper by summarizing the results from Guez et al. (2019) and added a contributions subsection in the introduction. We have also reformatted the paper to clearly specify the 4 different hypothesis we have in section 3 and explained before each section which hypothesis we are testing there.
>
> > In Section 4 on linear probing the predictive power of the next box and next target probes are much lower than direction probes. I think intuitively should these not be a much better measure of planning depth as they are more relevant to long term plans?
>
> Note that we use linear models as probes. The comparatively low F1 score for next target/box probe shows that the information for the next target is not exactly linearly encoded in the hidden states. One reason could be that the agent directions indirectly encodes the next target information which should be extractable using a non-linear probe. We have added a comment clarifying this in the paper.
>
> With that said, the NN does not necessarily need to encode the next boxes/targets in a local, linear way.
>
> > Section 5 shows some generalisation ability of DRCs but this has already been shown in a variety of settings by Guez et al. (2019) and I think it is not super clear what the current paper adds.
>
> Guez et al. (2019) showed the generalization to levels with more boxes but the same grid size of, $10 \times 10$. However, our results are more striking since we check generalization to much larger levels with many boxes as shown in Figure 2. In fact, we do this by removing the MLP layer and using the next action information linearly encoded in the last layer’s hidden state channels. We show that the underlying algorithm mechanism that's learned is generalisable even when the network naively isn't
>
> > the section on linear probes finds good predictiveness but much worse causal probe results. Why is this the case?
>
> Our probe results show that many of the probes are highly predictive but only the box directions probe is highly causal to modify the agent’s next action. This shows that the future box directions computation is part of the model’s algorithm to predict action. A probe trained to predict a property $P$ that has high predictive score but low causal score shows that the information about $P$ can be read from the activations but $P$ is not part of the model’s algorithm to predict action. It could be that $P$ is used by the network to do something else (e.g., predict the value of the state) or $P$ correlates with the activations. For example, we found that the agent directions probe is always encoded (highly predictive) but is only causal when the box directions representations are **not** informative for predicting the next action, as shown in the case study of section 4.3. This results in lower causal scores for the agent directions probe. We have made this interpretation of the causal probes results more clear in the paper.
>
> > it is not clear to me how the presented work connects to the “Ethical treatment of AIs” or to “Goal misgeneralization and mesa-optimizers for alignment”.
>
> Regarding ethical treatment of AIs related work, we believe it is important and related and we have explained it better now. However, several reviewers had raised that this is a strange argument, so we have happily moved it to the appendix.
>
>
> > What is special about pacing as opposed to other forms of moving around?
>
> The pacing in cycle behavior is a completely redundant behavior that an optimal policy (such as that computed using A*) would not perform. However, this behavior was implicitly learned by RL to provide DRC more compute during inference to come up with better plans as shown in the paper. This is clearly distinct from other forms of moving around such as moving in a straight line or pushing a box which are goal-directed and are based on the plans that DRC forms early on.
>
> We have added new evidence in lines 399-416 of the updated paper to show that cyclic-pacing behavior is more special. You can find a summary of these new evidence in the new common response.
>
> > What is being aggregated over in Section 5.1.?
>
> The action probes are applied across channels of the last layer’s hidden state $h$ for individual squares in the grid, which results in a $10 \times 10$ probe output for each action. We aggregate this output using the spatial-aggregation methods in section 5.1. We have clarified this in the updated paper.

---

### Official Review · Reviewer_9qTN · 2024-11-03

**Soundness:** 1
**Presentation:** 1
**Contribution:** 1
**Rating:** 3
**Confidence:** 3

**Summary:**

This paper makes an study on how a neural network within an RL agent learns to play Sokoban. Specifically, this work studies how and where the network abstracts the problem at hand, plans and where decision making takes place.

**Strengths:**

The paper explores an interesting research direction on how a neural network within an emobied agent abstract their problem at hand, projects a plan and makes the decision making. It is an interesting problem that trhough a sound study can be very useful and benefitial for the community

**Weaknesses:**

While the line of research is of interest, the current paper has important shortcomings that require significant changes.

Specifically, there are three major weaknesses in this paper:

* Clarity: As I will detail below the paper is very difficult to understand and it would greatly benefit from a rewritting. Specifically, I would suggest authors to do a simple exercise at the beggining what are the hypothesis that are being tested, how this thesis is going to be tested and what results would be interpreted as a validation or violation of the hypothesis. Additionally, there are several concepts that are never introduced in the paper or that are introduced first and explained better. Last, a diagram about the architecture studied and where the probes (i.e., the system to track the hypothesis) are placed within the architecture would greatly help readers

* Generalization of the results: the whole study focuses exlusively on the environment of Sokoban and studies only two neural networks, casting doubts about how general the conclussions extracted are.

* Related literature. There are several works that study which drivers help best to RL agents to generalize in similar problems, both within the environment, the nn architecture or the perception field. The present paper doesn't follow any of the recomendations of those works,and they can be influential in the conclussions extracted, e.g. [1] demonstrated that for an RL agent to generalise and not memorize puzzle solving, should have an egocentric perspective, how does using a egocentric perspectice  or not effect planning?. [3,4,5] highlighted changes in the ability of rl agents to generalize their plans when using different architectures, do we observe changing on abstractions when using any of those configurations?


[1] Hill, Felix, et al. "Environmental drivers of systematicity and generalization in a situated agent." International Conference on Learning Representations.
[2] Hill, F., Tieleman, O., von Glehn, T., Wong, N., Merzic, H., & Clark, S. Grounded Language Learning Fast and Slow. In ICLR 2021.
[3] León, Borja G., Murray Shanahan, and Francesco Belardinelli. "Agent, do you see it now? systematic generalisation in deep reinforcement learning." ICLR Workshop on Agent Learning in Open-Endedness.
[4] León, Borja G., Murray Shanahan, and Francesco Belardinelli. "In a Nutshell, the Human Asked for This: Latent Goals for Following Temporal Specifications." ICLR 2022.
[5] Lake, Brenden M., and Marco Baroni. "Human-like systematic generalization through a meta-learning neural network." Nature 623.7985 (2023): 115-121.

**Questions:**

Besides my comments above some specific points on clarity:
* You never explain what probes are and how their work, if their reader is not already very familiar iwth them they won't understand the work.
* Lines 153-156 is difficult to follow the thinking process here, for instance you mention there that adding extra thinking steps solve 4.7% extra levels, but in line 157 you mention that the extra thinking  helps greatly in dificult problems. Are only a very small percentage of the results these har problems?
* It is not clear how the ResNet would perform cycles, the few it does
* Line 200-202: here is a good example of a point that a research question is made but the paper only offers evidence that "suggests" not that confirm. In that and the lines below i would be good to put down the reasoning that would give us certainty that the agent is pacying, and giving an intuition of what is pacing here.
*  The later two paragrapghs in section 4.0 would be better placed as a summary at the end of the section pointing out to where the claims are supported.
* Line 278 explain what is the standard of evidence in Li et al.

---

> ### Author Response · Authors · 2024-11-20
>
> Thank you for pointing out how the paper is unclear. We’ve definitely been thinking of the hypotheses and evidence for/against, but did not make that explicit enough. Hopefully the common response explains a bit better which hypotheses and evidence we are considering. (hypotheses: does this NN have plans? Does this NN do search?)
>
> Thank you also for the very thoughtful review – we’ve come up with better experiments about pacing that we will definitely incorporate into the next version of the paper.
>
> > the whole study focuses exclusively on the environment of Sokoban and studies only two neural networks
>
> In fact we only really study one network, the DRC. The ResNet is used as a baseline for performance and little else.
>
> Mechanistic interpretability requires manual work and is not easily repeatable for several architectures. But now that we have a methodology, we could re-run our spatial probe analysis in other grid environments (e.g. gridworld mazes) – would that be satisfactory for generalization?
>
> > There are several works that study which drivers help best to RL agents to generalize in similar problems
>
> Thank you for pointing out related works, we will definitely include them! Studying the different abstractions of the various architectures would be very interesting, if a bit difficult with the probing methodology.
>
> The focus of this paper is not on making better RL architectures or making RL architectures that reason more, but instead on studying RL architectures that are as close to standard as possible, and whether they plan. For this reason, we stayed as close to the normal RL pipeline as possible, though it is a bit unfortunate that we have this strange DRC architecture.
>
> ### Question Responses:
> 1. Thank you very much for mentioning this, we will explain in the paper: A probe is a simple machine learning model that predicts something based on the internal activations of a NN. It is meant to show where (if present) the NN represents certain concepts.
> 2. Thinking steps helps in solving an extra 4.7% levels on the medium difficulty validation set. On the hard difficulty set, the network solves extra 7.1% levels with thinking steps (see FIgure 1 Bottom left). Figure 3 (right) also shows increasing thinking steps helps in solving levels with longer optimal solutions.  Thank you for this question. We will clarify this on line 157 in the updated paper.
> 3. That’s correct. Since ResNet doesn’t have a recurrent state, any cycles that it may perform will always render the level unsolvable as it will keep moving in cycles. But it’s also disadvantageous for the DRC to do cycles!
> 4. Yes, good point. Here is our definition of pacing:
>     1. The agent takes more actions than are necessary to get to the next “state-altering” action. In this case: the path taken to the next box-push is longer than optimal
>     2. Because of these extra actions, the NN now ‘knows’ what actions to take, whereas before it did not.
> This suggests a very nice kind of experiment, on top of what we have: compare how accurate the probed plan is before and after the ‘on-policy’ actions of the NN, and compare them to just bee-lining to the next box. Thank you very much for prompting us to think of that!
> 5. Thank you for suggesting we put this summary at the end of the section. Is it because it’s difficult to understand without having read the section? Should we put them under a “Summary of evidence” header? We thought it would be better to tell the reader what to expect and what claims to look for, in advance, to help with skimming.
> 6. The standard of evidence is that the probes should be causal as well as predictive. We will add the explanation of the standard of evidence in the updated paper.

---

> > ### Comment · Reviewer_9qTN · 2024-11-24
> > **Follow up**
> >
> > I want to thank the authors for their response. I think the changes are moving in the right direction.
> >
> > - Mechanistic interpretability requires manual work and is not easily repeatable for several architectures. But now that we have a methodology, we could re-run our spatial probe analysis in other grid environments (e.g. gridworld mazes) – would that be satisfactory for generalization?
> >
> > To answer that let's go back to the question this paper addresses. "(hypotheses: does this NN have plans? Does this NN do search?)" If your goal is to respond this question only on the DRC, then clearly not, but the impact of this work would be limited, how do we know that there isn't anything particular with the DRC leaning to this results?
> >
> > Also if the hypothesis is to test  "does NNs have plans?" it should be defined what is a "having a plan" here, because one could say yes, the plan of the NN is to take the action that gets the best future value, that's how we wolve long-term games in RL. This study looks for something more complex than just getting the action of the best future value but that's why I believe this should be specified more carefully.
> >
> > Going back to generalization, I don't think that moving from sokoban to other grid like works tell us much unless the tasks on those other environments are very different. I would say probably miniworld, that is a 3D environment, could add something more on this regard.
> >
> > - Thank you for pointing out related works, we will definitely include them! Studying the different abstractions of the various architectures would be very interesting, if a bit difficult with the probing methodology.
> >
> > The focus of this paper is not on making better RL architectures or making RL architectures that reason more, but instead on studying RL architectures that are as close to standard as possible, and whether they plan. For this reason, we stayed as close to the normal RL pipeline as possible, though it is a bit unfortunate that we have this strange DRC architecture.
> >
> > Agree, but to study if RL architectures plan, wouldn't it be good to test if they always plan? or if they only plan under certain circumstances, and if they do when? It would be very simple for isntance reproduce a setting in the fashion of [1] which stated that a fixed perspective is very important for emergent properties in standard RL networks if that is found to have any effect on planning. Please understand that I am not stating you must cite or compare with that work, only that I believe that studying if planning and searching is observed differently under drivers that are know to make agents learn for instance in a compositional way, would strenghten the impact and interest of this work.
> >
> > Thank you for the detailed responses. Regarding "Thank you for suggesting we put this summary at the end of the section. Is it because it’s difficult to understand without having read the section? Should we put them under a “Summary of evidence” header? We thought it would be better to tell the reader what to expect and what claims to look for, in advance, to help with skimming."
> >
> > My problem while reading those paragraphs was that it was giving details before getting to understand what was it about and then there were things that were hard to find were where they in the text. Revisiting it now, for example I remember when reading this sentence "By the standard of evidence in Li
> > et al. (2023), we declare this conclusive evidence that the DRC(3, 3) represents and uses plans."  I wanted to dig more on where was that explained in the section, but there is no other place in Sec 4 that  Li
> > et al. (2023) is referenced. Thus, that intro confused me more until I did a couple of passes on the section, hence my recommendation about that might be better to give that info at the end.

---

### Official Review · Reviewer_iu25 · 2024-11-04

**Soundness:** 3
**Presentation:** 2
**Contribution:** 3
**Rating:** 5
**Confidence:** 3

**Summary:**

The paper dives into understanding the inner mechanisms of planning by deep reinforcement learning (RL) agents in the Sokoban game, especially using a deep repeated convLSTM (DRC) (Guez et al. 2019) network structure. This work proposes several interesting findings including the effect of thinking steps, "pacing", probing the interpretable information, and generalizability to different sizes of Sokoban. All these results unveil a path to open the black box of a (recurrent) neural networks for model-free planning. I believe the major contribution of the paper lies on, like the authors stated, understanding the inner objective of the planning network.

**Strengths:**

1. The current manuscript is highly informative a number of interesting analysis, many of which are novel to my knowledge (beyond those discussed in Guez et al. 2019).
2. Some of the results are thought-involking, which I have not thought about, such as the "pacing" mechanism.
3. The being touched task (planning in Sokoban) is quite interesting (while simple enough for humans to interpret) and potentially important for understanding planning.

**Weaknesses:**

While the current study presents a number of interesting results. I believe the manuscript could be improved in several aspects.

1. The paper could benefit from a better struture and more scientific style of writting. The current version, while informative, is not self-contained and often makes me feel difficult to understand the implementation details of each analyze method. The authors should assume minimal pre-knowledge of the methods used in this paper, for a more general audience.
2. The paper does spend enough efforts to convince the audience that the presented results are not cherry-picking. Although I tend to believe the presented findings are mostly general, the paper could be improved by e.g., discussing the variety among random seeds in training, and providing more error bars in the plots.

See the following questions for detailed comments.

**Questions:**

1. Could the authors clarify the relation to Guez et al. 2019 by explaining the overlapping parts between this paper and Guez et al. 2019, and new results.
2. For self-containness, could the authors provides a preliminary about DRC since it is the backbone of the current work.
3. Could the authors provides some illutrative examples  (rendering) of "pacing"?
4. How does the network deal with different size of input (other than 10x10) in the CNN part (Sec. 5.1)?  I am confused about how the convLSTM was adjust in Sec 5.1. Maybe a visualized diagram could help.
5. What was the learning objective in spatial aggregation ("We learn their relative weights ....... for 10000 steps.")?
6. DRC was compared only with ResNet, how about plain LSTM (as in Guez et al. 2019)? And I am curious about how a Transformer network performs in this case.
7. Many of the plots seems to miss error-bars, is it because the error bars are too small? Could the authors clarify the error bars (and number of repeats) not only in Tables but also in plotted curves?
8. In the Related Work section, could the authors make a more concrete explanation about "reasoning NN architectures"?
9. Is it necessary to include "ethical treatment of AIs" in the main texts?
10. Could the authors clarify "perhaps due to errors" ? (line 790)

---

> ### Author Response · Authors · 2024-11-20
>
> 1. We have provided a summary of the overlapping results from Guez et al. (2019) and the novel contributions we make through our paper in the common response. We are updating the introduction section in the paper to reflect this.
> 2. We describe the DRC architecture in slightly more detail in appendix A, but it’s pretty compressed. We will include a figure of the architecture in our updated paper.
> 3. You can find a video of the pacing behavior at the path `videos/pacing-cycles.mp4` in the uncompressed supplementary material. We are also adding a figure demonstrating pacing behavior in the paper.
> 4. The DRC architecture is based on a convolutional backbone with “same” padding such that it can take in any $M \times N \times 3$ image as input and process them into a $M \times N \times 32$ hidden states $h$ and $c$. Normally, this is flattened and fed into an MLP layer. In our model surgery section, we remove the MLP and spatially-aggregate the hidden state using mean-pooling, max-pooling, and proportion of positive squares readouts with action probes. This gives us 7 inputs as shown in Table 2 to convert hidden state to actions. We will add a visualization of the architecture in the paper with and without the model surgery.
> 5. We use the cross-entropy loss function to optimize the 7 parameters against the ground-truth actions taken by the network using the MLP layer. We include this in our updated paper.
> 6. We compare with the ResNet to demonstrate that the recurrent architecture is significantly more performative than bigger non-recurrent architectures. We agree with the reviewer that it is interesting to see how a transformer network performs in this environment and whether it will demonstrate a similar planning mechanism. We are planning to do that in the next project, the aim of this paper is to analyze the workings of the DRC architecture that Guez. et al (2019) found and described the behavior of.
> 7. Thank you for pointing this out, this is our mistake. Some figures have error bars: e.g. Figure 5(mid, right) and Figure 7. But for most figures we forgot, we focused on error bars for the tables. There are two kinds of error bars that make sense for our paper to have: error across seeds, and across dataset samples. Most of this paper is about interpreting a single trained NN (i.e. a single seed) so it only makes sense to add dataset error bars to figures: 3, 4, 5. Figure 1 would benefit from seed-based error bars because we’re making statements about training.
> 8. We’ll expand the section on “reasoning architectures” in the related work. It’s not a concrete category, we meant to highlight works that come up with NN architectures inspired by making them explicitly reason, or giving it a model of the world.
> 9. We believe it is important. However, several reviewers have raised that this is a strange argument, so we are happy to move it to the appendix.
> 10. We made bugs when setting the number of steps. It was important that the number of steps be divisible by 256x20, and the number on that line is divisible by that, but so is 2Billion (though not 1Billion). So basically: we wrote the wrong number before training, and after training we decided the number of steps was close enough to 2 billion that we would not bother running the training again.

---

> > ### Comment · Reviewer_iu25 · 2024-11-22
> >
> > Thanks for the response. I will wait for the revised manuscript and see.

---

### Author Response · Authors · 2024-11-20
**Common Response**

Thank you all for taking the time to evaluate our paper. We are delighted that all reviewers found our work novel. It is now clear to us that the paper is written badly. We take this very seriously, and aim to update the paper by Monday 25.

However, we stand by all the claims we make in this paper, and believe we have provided enough evidence for them. For the purpose of this discussion, we will summarize them again.

In the Introduction we define two features of goal-oriented algorithms:
- **Plan**: a sequence of actions the agent intends to take
- **Search algorithm:** an algorithm which considers many plans and picks the best one according to some evaluation of consequences.

We find very strong evidence that the NN has a **plan**, and weak (but still highly suggestive) evidence the model is doing **search**. (By ‘high suggestion’ we mean the likelihood p(observations | the model is doing online search) is high, see discussion with Reviewer cMKh)

**Plan**: Simple linear probes let us read out the plan and write it, in terms of box-pushes. This is the evidence we present for the NN having a *plan* and believe it is strong.

**Search**: The evidence for search is weaker because there are other possible explanations for why the model looks like it’s doing search (it has a ‘best plan’ it executes, more compute helps performance, training does not disincentivize taking steps for extra computation even though they impact returns negatively). But there aren’t very many concrete alternatives (e.g. various heuristics that still produce a plan, or the NN only plans at the beginning) so it is still suggestive evidence for search.


## How we will improve the writing

### Contribution statement

As reviewer p3dc points out, we need a better contribution statement that clearly delineates the work that we continue (Guez et al. 2019) and this paper. Guez et al.:
- Invented and described how to train the DRC architecture
- Showed that it solves many Sokoban levels, and solves even more when given ‘time to think’ with NOOPs
- Showed that it generalizes to 10x10 levels with up to 7 boxes, when trained exclusively on 10x10 levels with 4 boxes.

We replicate the NN and ‘thinking time with NOOP’ effects from Guez et al. The rest of our contributions are:
- **Strong evidence the NN computes a plan.** Using probes, we can read and write the NN’s activations in a way that predicts or influences its future behavior:
    - We train linear probes on the hidden state activations that are highly predictive of future agent and box moves, showing that DRC computes the future movements of boxes early on in a level.
    - We show that the box directions probe is highly causal in modifying the plan, while the agent directions probe is only causal when the box directions probe is not sufficient to inform the next actions.
        - Therefore the plan is internally represented as a sequence of box-pushes, not a sequence of actions.
    - Therefore the NN computes a plan.
- **Weak evidence the NN performs search.**
    - The DRC takes steps which do not advance the state of the game, despite it being penalized for extra steps in training.
    - This is fungible with extra thinking steps, suggesting that performing ‘useless actions’ does not perform extra functionality compared to NOOPs.
    - (from Guez et al.) the NN solves more levels when made to think with NOOPs
    - We show that improved performance from thinking time arises early in training and drops for the medium but not the hard levels (fig. 1 left). Also it is more helpful on longer-horizon levels (fig. 3). This shows it is more useful for harder levels
    - The NN generalizes very well to settings it was not trained for:
        - Levels larger than 10x10 and with N boxes
        - This is an extension of Guez et al.’s generalization to 7 boxes with 10x10 levels.
    - Together, these give us some evidence that the NN is doing search.



### Other points
1. We assume the knowledge of some terminology used in the interpretability community in the paper (probes, model surgery, model organism). We will update the paper to properly introduce these terms.
2. We will add a diagram with the DRC architecture in the appendix instead of just referring to Guez et al.
3. We will be updating the related works section by incorporating the suggested papers and establishing where our work fits in the literature.

We believe that the paper will become much stronger after we incorporate the writing feedback in the coming days. We look forward to the reviewer’s replies, now and after we update the paper.

---

### Author Response · Authors · 2024-11-29
**We have rewritten the paper**

We have updated the paper in the following ways based on the feedback from all the reviewers:

1. We have rewritten the introduction section by adding a paragraph summarizing the results from Guez et al. (2019). We also added a subsection, “1.1 Contributions,” describing all our contributions by delineating them based on evidence for plan and search.
2. We have a new figure describing the DRC(3, 3) architecture, along with visualizations of the different types of probes we apply and the model surgery we perform in fig. 2 (left).
3. We have a new section, “3. Hypothesis and Tools.” In 3.1, we describe our choice of tools and how we use them (probes and checking their causality). In 3.2, we provide definitions for 1. Plan, 2. Causal representations, 3. Search. We then describe 4 hypotheses we test in this paper: 1. The DRC(3, 3) has a plan, 2. Plan improve with computation, 3. Pacing to improve plans, 4. The NN is doing search.
4. For each section following section 3, we first describe the hypothesis this section tests.
5. In section 5.1, we added the following new results demonstrating strong evidence for hypothesis 3 based on the feedback from reviewer 9qTN (please refer to the paper for exact numbers and extra details):
  1. The F1 score of the future box-direction probe just before the DRC does cycles is smaller than when it doesn’t do cycles (at comparable timesteps in the episodes).
  2. The per-step increase in the F1 score is more than double for timesteps in cycles than in timesteps that are not in cycles).
  3. The per-step growth in the plan, as measured by new predictions of the box probe, is much higher for cycles than for non-cycles.
6. We have added new related work in the paper, discussing works along systematic generalization. We have moved some of the related work to the appendix.
7. We have added 95% confidence interval bars to all the plots in the paper. For some of these figures, the error bars are so small that you’ll have to zoom in on the paper to distinguish them.

We are incredibly grateful for the feedback from the reviewers and believe that the paper is much clearer and easier to follow now. We urge all the reviewers to retake a look at the paper and update their scores to reflect the new changes. The submitted version of the paper is 0.5 pages over the page limit. We have already cut down the paper to 10 pages by moving the less important results (such as the pacing probe) to the appendix, but we couldn’t submit it before the deadline.

We will be available to respond to any comments that the reviewers may have. We thank all the reviewers for their time and commitment to the review process!

---

### Meta-Review · Area_Chair_8QjV · 2024-12-20

**Metareview:**

The paper examines how a Deep Repeated ConvLSTM (DRC) neural network learns to play the game Sokoban, focusing on the emergence of planning behavior in this model-free agent. The authors claim the DRC exhibits "pacing" behavior, taking extra steps in complex scenarios to allow for more computation time. They further assert that linear probes can predict the network's future actions and that manipulating the hidden state through these probes allows for control over the agent's subsequent actions.

Strengths: The paper explores the under-researched area of planning in model-free agents. Specifically, it utilizes a variety of interpretability techniques to analyze the DRC network's internal representations and decision-making processes. Moreover, the use of Sokoban as the task domain is interesting, as it requires the agent to reason about sequences of actions and their consequences.  However, the original submission had several weaknesses:

- Lack of Clarity and Rigor: The paper uses vague language, poorly defined hypotheses, and an unstructured presentation that hinders comprehension. This lack of clarity extends to the methodology, which is not described in sufficient detail to allow for proper evaluation of the results.

- Insufficient Evidence and Limited Generalizability: The core claim of planning behavior is not convincingly supported by the evidence presented. The connection between the observed "pacing" behavior and actual planning is tenuous, and the limited experimental scope to Sokoban raises concerns about the generalizability of the findings.

- The paper fails to adequately contextualize its findings within the broader literature on planning in artificial intelligence. The related work section is inadequate and does not effectively position the research within the existing knowledge base.

Based on the discussion period, I'm leaning towards rejecting the paper despite author's attempts to address the above weaknesses, as they result in substantial modifications that seem to be appropriate for a revised submission to another conference.

**Additional Comments On Reviewer Discussion:**

During the rebuttal period, reviewers raised major concerns regarding the paper's clarity, methodology, and connection to the literature. Reviewer 9qTN pointed out the difficulty in understanding the paper and the need for a clearer presentation of the hypotheses and validation methods. Reviewer p3dc echoed these concerns and further questioned the clarity of the methods used in Section 5.1, the strength of the evidence supporting the claims, and the paper's overall connection to the relevant literature. In response, the authors attempted to address these concerns by adding a section on the hypotheses being tested and providing more details about the methods used. They also discussed the generalizability of their findings and attempted to clarify the connection to related work.

While the authors spent [some effort to address the reviewers' concerns]((https://openreview.net/forum?id=ORxjH9kTp8&noteId=ER89r3PIxA), the revision results in major modifications for addressing them, which seem beyond the scope of the current submission. Therefore, despite the authors' attempts to improve the paper during the rebuttal period, the revisions did not significantly alter my assessment of rejecting the paper.

---

### Decision · Program_Chairs · 2025-01-22

Reject